# Connexins Control Glial Inflammation in Various Neurological Diseases

**DOI:** 10.3390/ijms242316879

**Published:** 2023-11-28

**Authors:** Ryo Yamasaki

**Affiliations:** Department of Neurology, Neurological Institute, Graduate School of Medical Sciences, Kyushu University, Fukuoka 812-8582, Japan; yamasaki.ryo.510@m.kyushu-u.ac.jp; Tel.: +81-92-642-5340

**Keywords:** connexin, gap junction, hemichannel, astroglia, microglia, oligodendroglia, multiple sclerosis, autism

## Abstract

Connexins (Cxs) form gap junctions through homotypic/heterotypic oligomerization. Cxs are initially synthesized in the endoplasmic reticulum, then assembled as hexamers in the Golgi apparatus before being integrated into the cell membrane as hemichannels. These hemichannels remain closed until they combine to create gap junctions, directly connecting neighboring cells. Changes in the intracellular or extracellular environment are believed to trigger the opening of hemichannels, creating a passage between the inside and outside of the cell. The size of the channel pore depends on the Cx isoform and cellular context-specific effects such as posttranslational modifications. Hemichannels allow various bioactive molecules, under ~1 kDa, to move in and out of the host cell in the direction of the electrochemical gradient. In this review, we explore the fundamental roles of Cxs and their clinical implications in various neurological dysfunctions, including hereditary diseases, ischemic brain disorders, degenerative conditions, demyelinating disorders, and psychiatric illnesses. The influence of Cxs on the pathomechanisms of different neurological disorders varies depending on the circumstances. Hemichannels are hypothesized to contribute to proinflammatory effects by releasing ATP, adenosine, glutamate, and other bioactive molecules, leading to neuroglial inflammation. Modulating Cxs’ hemichannels has emerged as a promising therapeutic approach.

## 1. Introduction

Gap junctions, assembled from connexins (Cxs), are crucial for direct cell-to-cell communication [1]. The exploration of gap junctions began with their discovery by Revel and Karnovsky [2], and that of cloning began with Kumar and Giulia [3], and Paul [4], followed by the cloning and sequencing of connexin 43 (Cx43), a 43-kilodalton (kDa) protein, by Beyer [5] and Cx26 by Zhang and Nicholson [6]. Humans have a family of 21 Cx genes, and most cell types express multiple Cx isoforms. Cxs are named based on their molecular weight or structural characteristics, with agreed-upon nomenclature (see Table 1) [7]. Six Cxs form connexons or “hemichannels,” which can assemble in homotypic or heterotypic combinations. Two connexons align head-to-head to create gap junctions between neighboring cells, bridging the gap homotypical or heterotypical (see Figure 1) [1]. Notably, Cxs have a relatively short half-life of 1 to 5 h [8,9], emphasizing the need for rapid adjustments in gap junction-dependent intercellular communication. Therefore, most cells renew their gap junctions within 24 h. The life cycle of Cxs involves translation, insertion into the endoplasmic reticulum (ER), oligomerization into homomeric or heteromeric single-membrane channels, and transport to the Golgi apparatus. Upon delivery to the cell surface, Cxs function as hemichannels or form gap junction channels, clustering and assembling into functional gap junctions with neighboring cells. After a brief period at the cell surface, gap junction fragments are internalized as double-membraned connexosomes and eventually degraded in lysosomes [10]. This dynamic regulation of Cxs underscores their critical role in facilitating intercellular communication.

Multiple sclerosis (MS) is a demyelinating disease that affects the central nervous system (CNS). It has profound implications on the intricate CNS landscape. MS lesions involve a complex interplay between infiltrating inflammatory cells, demyelinated neurons, and activated glial cells. Within this environment, activated microglial cells clear cellular debris, and activated astroglial cells encircle the lesion site [11]. Notably, even in chronic MS lesions, where peripheral immune cells and myelin-laden microglia are absent, astroglial cell activation, also known as astrogliosis, persists [12]. This enduring astroglial activation suggests they play crucial yet enigmatic roles in both acute and chronic phases of MS.

Astroglia are a primary glial cell type in the CNS, alongside oligodendroglia and microglia. Under normal conditions, astroglial functions are multifaceted. They offer mechanical support to the neuronal network and vital biological support, including K^+^ buffering, H^+^/Ca^2+^ regulation, neurotransmitter uptake (e.g., glutamate and GABA), the control of cerebral blood flow, water transport, oxidant neutralization, and glucose transport [13]. Additionally, astroglia promote myelination by producing growth factors that induce the differentiation of oligodendroglia progenitor cells [14]. These functions rely on the coordinated actions of glial cells connected via gap junctions. In the complex realm of MS, where ongoing discoveries about pathomechanisms are made, the roles of astroglial cells and Cxs are crucial. In acute MS lesions, characterized by infiltrating immune cells and demyelinated neurons, and in chronic lesions with a subsiding inflammatory environment but persistent astroglial activation, these components play pivotal yet mysterious roles. Their multifaceted functions include vital support, precise regulation, and intricate communication, providing a promising avenue for understanding MS pathogenesis and developing potential therapies. Dedicated scientists persist in unraveling the mysteries surrounding the roles of astroglia and Cxs, yielding novel insights with the potential for advanced treatments and a deeper understanding of the intricacies of CNS functioning in both health and disease.

## 2. Connexins in the Nervous System

### 2.1. Connexins in the Peripheral Nervous System

Cx32 is expressed by Schwann cells and is thought to be important for the proliferation of Schwann cells, re-myelination, and nerve regeneration [15]. The location of Cx32 expression indicates a radial diffusion pathway between adjacent layers of non-compacted myelin for signaling molecules, metabolites, or ions from the inner (adaxonal) to the outer (abaxonal) cytoplasm. Cx29 is also expressed on myelinating Schwann cells and is reported to work as a hemichannel [16].

### 2.2. Connexins in the Central Nervous System

Oligodendroglia in the CNS express Cx29, Cx32, and Cx47 [17]. Cx32 and Cx47 are the components of homotypic (Cx32–Cx32; Cx47–Cx47) gap junctions but are unable to assemble Cx32–Cx47 heterotypic gap junctions [18]. Astroglia express Cx26, Cx30, and Cx43. Cx26 and Cx30 form homotypic and heterotypic gap junctions, but Cx43 only forms homotypic gap junctions and never forms heterotypic coupling with Cx26 and Cx30. Meanwhile, Cx43 forms heterotypic gap junctions with oligodendroglial Cx47 (Cx43–Cx47). Astroglial Cx26 and Cx30 also form heterotypic gap junctions with oligodendroglial Cx32 (Cx30–Cx32; Cx26–Cx32) [18]. These astroglia–astroglia, oligodendroglia–oligodendroglia, and astroglia–oligodendroglia couplings are important for the propagation of Ca^2+^ waves, transfer of metabolites, and spatial buffering of K^+^ or neurotransmitters as a result of neuroaxonal or synaptic activity [18].

Neurons express Cx36. Cx36 assembles homotypic gap junctions to contribute to electrical synapses [19]. Neurons and glia are typically not directly connected by gap junctions [20]. However, there have been rare reports of gap junctions between demyelinated naked axons and astroglia, as documented by Soffer in 1980 [21]. Microglia, the only innate immune cells that originate from the myeloid precursor cell lineage, are known to express Cx32 and Cx36 in the resting stage, while Cx29, Cx32, Cx36, and Cx43 expression levels are upregulated in the activated stage [22]. Cx43 is proposed to form hemichannels to release ATP and glutamate, and allow Ca^2+^ influx [1,23]. Another important function of microglia is to modulate Cx expression on other glial cells. Cx43 levels and dye coupling are reduced in astroglia when co-cultured with microglia. Interleukin-1β (IL-1β) and tumor necrosis factor-α from activated microglia reduce the Cx43 levels on astroglia. Interferon gamma released from Th1 cells was suggested to induce microglial activation [24]. IL-1β has been reported to enhance calcium wave activity in primary-cultured astroglia through P2 receptors and reduce the expression of Cx43 on the cell surface [25]. Therefore, the microglia-induced indirect activation of neighboring astroglia is likely mediated by inflammatory cytokines [24,25]. Microglia were reported to modulate Cx43 gap junction-mediated dye coupling [26]. The decrease in gap junction-forming Cx43 and increase in Cx43 hemichannels on astroglia by inflammatory cytokines also may potentiate chronic brain inflammation [27]. Indeed, De Bock et al. have shown that Cx43 hemichannel opening in astroglia was induced by IL-6 in a calcium-dependent manner and that LPS-triggered astrogliosis could be prevented by TAT-Gap19, a selective Cx43 hemichannel inhibitor, or BAPTA-AM, which suppressed Cx43 hemichannel opening [28].

## 3. Connexins and CNS Glial Inflammation in Demyelinating Diseases

In MS lesions, expression levels of astroglial Cx43 and oligodendroglial Cx47 are decreased. In the chronic phase, however, Cx43 expression is upregulated, while Cx47 expression remains downregulated [12]. As mentioned above, inflammatory cytokines, such as IL-1β and TNF-α might suppress the expression levels of Cxs in the acute phase with the massive infiltration of peripheral immune cells in acute lesions. In chronic lesions, peripheral immune cells are sparse, with massive astrogliosis. These observations indicate an imbalance of Cx43 and Cx47 in chronic lesions that leads to an increase in Cx43 hemichannels [12]. We recently tried to elucidate the pathomechanisms of chronic MS using an animal model of the disease, which is known as experimental autoimmune encephalomyelitis (EAE). A study reported that in oligodendroglial Cx32-deficient mice, the clinical course of EAE was aggravated with failed re-myelination. Cx32 is expressed on the peripheral myelin; therefore, disrupted re-myelination might be induced by metabolic failure [29]. In this section, we will introduce several studies that tried to elucidate functional role of various Cxs/pannexin including our results.

### 3.1. EAE in Whole Body Cx30-Deficient Mice

Cx30, primarily expressed by activated astroglia, exhibits increased expression during EAE induction in wild-type mice. Notably, EAE induction in Cx30-deficient mice alleviates clinical signs specifically in the chronic phase [30]. Despite mild clinical signs, microglia are activated, with no impact on the infiltration of CD3^+^ T cells. There is a ramified microglia increase, displaying enhanced anti-inflammatory characteristics marked by arginase-1 and brain-derived neurotrophic factor upregulation, and nitric oxide synthase 2 downregulation during both acute and chronic EAE phases. In the absence of EAE, Cx30 deficiency results in slightly enlarged astroglial processes in the spinal cord’s gray matter and a partial reduction in Cx43 expression in the white matter. During acute EAE, astroglia in Cx30 knockout (KO) mice exhibit earlier and stronger activation, showing an increased expression of ‘A2’ astroglia markers [31] and a significant decrease in Cx43 during the chronic phase. In the chronic phase of EAE, spinal cord neurons and axons exhibit better preservation in Cx30 KO mice compared to their littermates. These findings suggest that Cx30 deficiency promotes the presence of ramified microglia in the CNS in the absence of disease and improves chronic EAE by steering microglia towards an anti-inflammatory phenotype, highlighting an unknown important role of astroglial Cx30 in regulating the number and functional state of microglia [30].

### 3.2. EAE in GLAST-Positive Astroglia-Specific Cx43-Ablated Mice

The whole-body Cx43 knockout mouse is lethal [32]; therefore, we ablated Cx43 only in astroglia. Astroglia have several phenotypes. We chose glial fibrillary acidic protein (GFAP) and glutamate-aspartate transporter (GLAST) as astroglia-specific molecular markers. GFAP-positive astroglia exist on white matter, while GLAST-positive astroglia reside in the brain cortex [33,34]. When we induced EAE in GLAST-positive astroglia-specific Cx43 ablated mice, the onset was almost the same as that in the wild-type control, but the severity was significantly ameliorated from the acute phase, and a mild clinical course was sustained until the chronic phase. In spinal cord lesions, inflammatory cell infiltration was decreased, with reduced demyelinating areas. Because the chronic EAE signs are determined by the initial infiltration of inflammatory cells, the main cause of the mitigation of EAE signs is the inhibition of the initial infiltration of immune cells. We also found that in the genetically modified mice, inflammatory cytokine levels in the cerebrospinal fluid (CSF) at the pre-onset phase were significantly decreased compared with those in control mice. The GLAST protein is not expressed by spinal cord astroglia; therefore, the remote effects of Cx43 ablation on brain cortical astroglia are likely to be caused by the alleviation of CSF inflammation [35].

### 3.3. EAE in Cx30 Null/GFAP Positive Astroglia-Specific Cx43 Ablated Mice

Lutz et al. reported that Cx30-null/GFAP-positive astroglia-specific Cx43-ablated mice exhibited a vacuolation of white matter oligodendroglia and myelin edema [36]. They anticipated the aggravation of EAE in this model, but there were neither changes in the clinical/pathological expression of acute EAE nor an increase in BBB permeability [37]. Furthermore, we could not detect any significant difference in EAE clinical severity in Cx30-deficient mice in the acute phase but found a mitigation of clinical, pathological, and genetic phenotypes in the chronic phase of EAE. The reason for the differing results between the GLAST-positive astroglia-specific ablation of Cx43 and the GFAP-positive astroglia-specific ablation of Cx43 needs further clarification.

### 3.4. EAE in Oligodendroglia-Specific Cx47 Ablated Mice

We induced EAE in oligodendroglia-specific Cx47-ablated mice. Surprisingly, EAE became significantly worse through the clinical course, from the acute phase to the chronic phase, in contrast to the phenotype of the astroglia-specific Cx-ablated mice. In spinal cord lesions of Cx47-ablated mice, the infiltration of Th17 cells was increased. RNA array assays of microglia isolated from these mice showed inflammatory phenotypes with injury-responsive gene expression signatures. These activated microglia also expressed chemokine-related genes such as *Ccl2*, *Ccl3*, *Ccl4*, *Ccl7*, and *Ccl8*. Astroglia-specific gene signatures from brain homogenate showed inflammatory A1-deviated phenotypes [38].

Following our experiments, Stavropoulos et al. published an excellent paper revealing the exacerbation of EAE in Cx47-deficient mice. The study demonstrated an altered expression of *Vcam-1*, *Ccl2*, and *Gm-Csf* genes. Additionally, they identified blood–spinal cord barrier dysfunction characterized by defects in tight junction formation and increased T cell infiltration before the onset of the disease. Their data suggested the involvement of CNS cell populations expressing Cx47, beyond oligodendroglia, such as the CNS lymphatic epithelium [39].

### 3.5. EAE in Pannexin1 Deficient Mice

Pannexin1 (Panx1) is a cell membrane channel that allows the passage of relatively large molecules, such as ATP. In the CNS, Panx1 is found in neurons and glial cells, and in the immune system, it is present in macrophages and T cells [40]. Lutz et al. conducted a study to test the hypothesis that Panx1-mediated ATP release contributes to the development of EAE, using both wild-type (WT) and Panx1 KO mice [41]. The results showed that Panx1 KO mice exhibited a delayed onset of clinical signs of EAE and a reduced mortality rate compared to WT mice, although they developed comparable clinical severities to the WT mice. Additionally, during the acute phase of the disease, Panx1 KO EAE mice had fewer spinal cord inflammatory lesions. Furthermore, when they used mefloquine (MFQ), a pharmacological inhibitor of Panx1 channels, it reduced the severity of both acute and chronic EAE, whether administered before or after the onset of clinical signs. ATP release and YoPro uptake were significantly increased in WT mice with EAE compared to those in non-EAE WT mice, whereas these were reduced in the tissues of Panx1 KO EAE mice. They also observed an upregulation of the P2x7 receptor in the chronic phase of EAE in both WT and Panx1 KO spinal cords. This increase in receptor expression likely compensates for the decrease in ATP release observed in Panx1 KO mice and contributes to the development of EAE symptoms in these mice. This study demonstrates that a Panx1-dependent mechanism involving ATP release and/or inflammasome activation plays a role in disease progression, and inhibiting Panx1 through pharmacological means or genetic disruption delays and mitigates the clinical signs of EAE [41].

### 3.6. Tuning of the Brain Inflammatory Milieu via the Alteration of Connexin Channel Functions

In the above animal models of EAE with the functional modification of gap junctions on glial cells (Table 2), we can speculate the roles of the Cx hemichannels as the immuno-modulators in the pathomechanisms of both acute and chronic phases of EAE. First, the modification of gap junction functions affects the pathomechanisms of demyelinating inflammatory CNS disorders, not only by altering myelin metabolism, but also by affecting inflammatory cell infiltration. Second, it is hypothesized that an imbalance of astroglial Cx43 and oligodendroglial Cx47 caused the aggravation of EAE via the relative increase in Cx43 hemichannels, which leads to the increase in extracellular bioactive molecules such as ATP or glutamate, resulting in the upregulation of CSF cytokine levels, pro-inflammatory activation of microglia, and increased infiltration of peripheral immune cells [38]. Cxs are expressed by all CNS cells and assemble a direct glial network. We hypothesized that when one gap junction-forming Cx is ablated, other Cxs might become closed hemichannels. The opening and closing of hemichannels depend on the intra- and extra-cellular milieu. The initiation of autoimmune demyelinating CNS disease occurs via the infiltration of autoreactive T cells. These T cells release interferons and other pro-inflammatory cytokines. These cytokines not only directly promote the chemotaxis of peripheral immune cells but also activate astroglia and facilitate hemichannel opening. Activated microglia release inflammatory cytokines to further activate microglia. Consequently, peripheral and glial inflammation is accelerated. After the acute phase of the disease, regulatory T cell infiltration ceases the activation of peripheral immune cells, but glial inflammation is sustained, which leads to progressive CNS damage in the chronic phases (Figure 2). Regarding Cx imbalance and CNS inflammation, De Bock et al. recently demonstrated that LPS and proinflammatory cytokines trigger endothelial hemichannel opening while IL-6 activates astrocytic hemichannels; thus, targeting Cx hemichannels is a promising therapeutic approach to suppress barrier leakage as well as astrogliosis [28]. Delvaeye et al. have previously shown that blocking Cx43 hemichannels with TAT-Gap19 protected mice against TNF-induced mortality, hypothermia and vascular leakage in the course of an acute cytokine storm [42]. As mentioned in the excellent studies from other groups [29,37,39,41], the role of Cxs extends beyond immune regulation and is deeply involved in the metabolism and cell repair mechanisms of many CNS cells, collectively shaping the pathophysiology of EAE and beyond (Table 2).

### 3.7. Targeting Glial Inflammation to Treat the Chronic Phase in the Autoimmune Demyelinating Disease Model

As activated microglia are largely responsible for chronic demyelination, we tried to treat EAE mice with anti-inflammatory drugs, which are available commercially as disease-modifying anti-rheumatic drugs (DMARDs), such as iguratimod (IGU). IGU has a suppressive effect on the activation of peripheral immune cells like T cells, B cells, and macrophages. When we orally treated EAE mice with preventive IGU (before the onset of EAE), the onset of EAE was almost perfectly precluded. Even with treatable administration (when treatment started after the peak phase of EAE), the clinical severity was significantly reduced. In vitro assays revealed a direct suppressive effect on isolated microglia [43,44]. Because IGU is already commercially available and has been shown to be safe with minimal adverse effects, it can be considered a promising candidate for treating autoimmune neuroinflammatory diseases such as MS or neuromyelitis optica. Overall, IGU demonstrates anti-inflammatory properties that alleviate disease symptoms not only by reducing peripheral inflammation but also by inhibiting immune cell migration and attenuating glial inflammation within the affected areas.

Another promising candidate drug as a hemichannel blocker, INI-0602, which is one of the derivatives of carbenoxolone, is currently under investigation for its anti-inflammatory effects on EAE. INI-0602 is a centrally acting, pan-Cx inhibitor with a higher affinity for hemichannels than for gap junctions [45]. It selectively blocks the release of small molecules such as glutamate from microglia through hemichannels, without attenuating the induction of acute inflammatory cytokines. Previous reports have shown that inhibiting this pathway with INI-0602 had no effect on inflammatory cytokines, yet it suppressed disease progression in mouse models of amyotrophic lateral sclerosis (ALS) and Alzheimer’s disease (AD) by blocking the release of glutamate into the extracellular space [45]. Importantly, INI-0602 was designed to target the CNS. The CNS redox system oxidizes the dihyropyridine moiety of the pro-drug, resulting in the drug being trapped within the CNS. Therefore, while INI-0602 is rapidly cleared from the circulation and peripheral tissues, it accumulates in the regulated CNS region. This pharmacodynamic property of INI-0602 has been reported to result in no systemic side effects in mice even after chronic administration for 5 months [45].

## 4. Connexins Involved in Other Neuronal Diseases

### 4.1. Heritable Diseases Caused by Connexin Gene Mutations

Congenital Cx protein alteration might lead to a misplacement of connexons within the cell or, in the case of recessive mutations, the inability to create connexons (either of the same type or the correct mixed type). Alternatively, some Cx protein alterations may permit the formation of mutant connexons but prevent the creation of gap junctions with connexons from other cells. Alternatively, certain alterations may enable the formation of complete gap junctions between cells, but they may significantly change properties like conductivity, molecular passage, phosphorylation, and voltage control, ultimately affecting cellular function in either a dominant negative or gain-of-function manner. Moreover, Cx mutations can cause a gain-of-function effect by increasing the permeability of hemichannels, leading to “leaky” hemichannels and cell death [46].

#### 4.1.1. X-Linked Charcot–Marie–Tooth Disease (CMTX1) (OMIM 302800)

Charcot–Marie–Tooth (CMT) disease is a group of inherited peripheral nervous system diseases. Charcot–Marie–Tooth disease, X-linked 1 (CMTX1), is caused by a mutation in Cx32. The clinical course is slowly progressive sensorimotor neuropathy with subclinical abnormalities of visual and brainstem auditory evoked responses, CNS white matter abnormalities, pyramidal tract dysfunctions, and deafness. The disease results from malfunctions of Schwann cells. While the localization and trafficking of the mutant protein in cell culture was normal, electrophysiological studies revealed that the involved Cx32 missense mutation F235C caused aberrant hemichannel opening with excessive membrane permeability and adverse effects on Schwann cell viability [47,48].

#### 4.1.2. Oculodentodigital Dysplasia (ODDD) (OMIM 164200)

ODDD is an autosomal dominant syndrome caused by *GJA1* gene mutations, and causes Cx43 malfunctions [49]. The symptoms of ODDD can vary widely among individuals and may affect the eyes, teeth, fingers, and other body parts. Common features include abnormalities in the fingers and toes, dental problems such as small teeth and early tooth loss, and vision issues. Facial dysmorphism, such as a thin nose and small mouth, can also be present. Brain magnetic resonance imaging (MRI) shows a hypointensity of gray matter and hyperintensity of occipital and periventricular white matter. ODDD mutations affect Cx43 trafficking as well as channel functions [50]. The ODDD Cx43 mutants G138R and G60S have been characterized to display increased hemichannel function combined with decreased gap junction function. The I130T mutant has been linked to ventricular tachyarrhythmia in a mouse model [51].

#### 4.1.3. Pelizaeus–Merzbacher-Like Disease (PMLD1) (OMIM 608804)

PMLD1 cases that show symptoms indistinguishable from those of Pelizaeus–Merzbacher disease (PMD) but do not show abnormalities in the *PLP1* gene are called PMLD and are distinguished from PMD [52]. It is a rare disease and has an autosomal recessive inheritance pattern. Some PMLD1 cases are caused by a mutation in the gap junction protein C2 (*GJC2*) (*Cx47*) gene. Upon MRI, the pyramidal tract is relatively spared compared with that in PMD; therefore, motor development tends to be better than that in PMD, including acquiring a sitting position and walking alone, but regression begins in the teenage years. Currently, there is no effective treatment; there is only symptomatic treatment.

#### 4.1.4. Nonsyndromic Hearing Loss and Deafness (DFNA3) (OMIM 601544)

DFNA3 is a subtype of nonsyndromic hearing loss that is inherited in an autosomal-dominant pattern. It is caused by mutations in the *GJB2* gene, which produces Cx26. Cx26 is important for the functioning of gap junctions in the inner ear. Mutations in the *GJB2* gene disrupt the formation or functioning of Cx26, leading to impaired communication between cells in the auditory system and resulting in progressive, sensorineural hearing loss. DFNA3 is characterized by a gradual loss of hearing, and the severity and age of onset can vary. Diagnosis involves clinical evaluation, family history assessment, and genetic testing to identify *GJB2* gene mutations. Management typically involves hearing aids or other assistive devices, and in severe cases, cochlear implants may be recommended. Hearing loss in syndromic deafness likely differs from the nonsyndromic form with mutant hemichannels that are overactive and/or show altered Ca^2+^ permeability leading to cell dysfunction and death [53]. It is also well known that a single mutation, 35delG, is responsible for the majority of cases of autosomal recessive hearing loss known as DFNB1. This specific gene mutation stands as the most prevalent cause of hearing loss within American and European populations, boasting a carrier rate of approximately 3%. Additionally, the mutation 167delG, another deletion variant causing DFNB1, holds a carrier rate of around 4% within the Ashkenazi Jewish population [54].

#### 4.1.5. Keratitis–Ichthyosis–Deafness (KID) Syndrome (OMIM 148210)

Keratitis–ichthyosis–deafness (KID) syndrome is a rare genetic disorder that affects the skin, eyes and ears [55,56]. It is also caused by mutations in the *GJB2* gene, which encodes Cx26 [57]. The skin manifestations of KID syndrome include the thickening and scaling of the skin, which resembles a condition called ichthyosis. These skin abnormalities can cause significant discomfort and may lead to complications such as infections. KID syndrome also affects the eyes, resulting in a condition called keratitis. Additionally, individuals with KID syndrome often experience hearing loss, which can be present from birth or develop over time. The hearing loss in KID syndrome is typically sensorineural, affecting the ability to perceive sounds [58]. The management of KID syndrome focuses on treating the specific symptoms and may involve the use of medications to alleviate skin and eye manifestations, hearing aids or cochlear implants for hearing loss, and supportive care [59]. Levit et al. reviewed the role of hemichannels in the pathogenesis of KID syndrome [60]. Overactive, “leaky” hemichannels may deplete the cytoplasm of essential metabolites, depolarize the plasma membrane, induce apoptosis via Ca^2+^ overload, or cause lysis via osmotic imbalance.

### 4.2. Seizures and Epilepsy

Epileptogenicity is usually caused by the propagation of abnormal discharges through synaptic junctions, but some studies have investigated the role of gap junctions. Cx43, Cx36, and Panx1 are among the Cxs that have been studied in relation to epilepsy. Each of these Cxs may play a specific role in the context of epilepsy.

Cx43 is one of the most extensively studied Cxs in epilepsy research. Altered expression and function of Cx43 have been associated with various forms of epilepsy, including temporal lobe epilepsy [61]. Cx43 has been implicated in the spread of seizure activity and the modulation of gliotransmission, which can influence neuronal excitability [62]. The administration of the selective Cx43 hemichannel inhibitor TAT-Gap19 has been shown to decrease seizure activity in animal epilepsy models, indicating the involvement of Cx43 hemichannels in epilepsy [63].

Cx36 is primarily found in neurons and is involved in forming electrical synapses, particularly in inhibitory interneurons. Changes in Cx36 expression or function have been linked to epileptiform activity in hippocampal CA1 and CA3 [64]. It is thought to contribute to network synchronization, which can be both pro-epileptic and anti-epileptic depending on the context. In slice cultures of the Cx36-deficient mouse hippocampus, 4-aminopyridine-induced seizure activity was reduced compared with that in slices from wild-type mice [65]. In addition, Cx channel blockers successfully showed anti-epileptic effects in both slice cultures and in in vivo models [66].

Panx1 is a protein that forms pannexon, also known as pannexin 1 channel [67]. It is expressed in various cell types, including neurons and glial cells. Panx1 channels can release ATP, which acts as a signaling molecule in the brain [68,69]. Alterations in Panx1 function have been associated with increased neuronal excitability and susceptibility to seizures [70]. Panx1 has been studied in the context of both epileptogenesis and seizure propagation [71].

Cx32 deficiency leads to neuronal hyperexcitability. A study examined the impact of Cx32 deficiency on the structure and function of the neocortex in adult mice. Morphological analysis revealed a reduced myelin volume and thinner axonal myelin sheaths in Cx32-deficient mice. Electrophysiological recordings indicated increased membrane resistance and enhanced intrinsic excitability in neurons of Cx32-null mice. Additionally, a significant portion of these neurons exhibited delayed and large glutamatergic excitatory postsynaptic potentials, reminiscent of paroxysmal depolarizations, with deficient GABAergic inhibition. The observed changes in intrinsic membrane properties were independent of alterations in synaptic function. Overall, Cx32 deficiency in the neocortex is associated with myelination defects, intrinsic membrane property changes, and impaired inhibitory synaptic transmission. These findings indicate the importance of gap junctions in the pathomechanisms of epilepsy [72].

### 4.3. Brain Ischemia

Ischemic brain injury induces the disruption of the blood–brain barrier and leads to the robust infiltration of peripheral immune cells and plasma, which results in drastic changes to the environment. Usually, Cx hemichannels are closed, but they can open under pathological conditions [73]. Several agents that block gap junctions appear to be neuroprotective, whereas, in genetic models, a reduction in or deletion of Cx expression leads to an exacerbation of clinical signs. Carbenoxolone, a gap junction blocker, reduces lipid-peroxide formation, which may exert neuroprotective effects [74]. In addition, knocking down specific Cxs, like Cx32 and Cx26 in neurons or Cx43 in glial cells, provided significant neuroprotection. This suggests that gap junctional communication contributes to ischemic injury propagation (so called “bystander effects”), and targeting specific gap junctions may reduce brain damage [75]. Freitas-Andrade et al. have demonstrated that the pharmacological blockade of Cx43 hemichannels with TAT-Gap19 significantly decreased infarct volume in an animal stroke model, and the detrimental hemichannel activity following ischemic stroke was linked to MAPK phosphorylation of sites at the Cx43 C-terminus [76]. Chen et al. found that Gap19 exhibited neuroprotective effects on cerebral ischemia/reperfusion and that Gap19 reduced the inflammatory response via the inhibition of the TLR4 signaling pathway following in vivo middle cerebral artery occlusion and in vitro oxygen glucose deprivation [77]. On the contrary, Naus et al. reported that blocking gap junctions during a glutamate insult to co-cultures of astroglia and neurons results in increased neuronal injury. They hypothesized that gap junctions play a neuroprotective role against glutamate toxicity [78]. The effects of blocking gap junctions in the context of an ischemic insult may vary depending on the pathological circumstances.

### 4.4. Motor Neuron Disease

Neuronal synapses comprise not only the metabolic synapses but also electrical synapses [79]. We observed reduced numbers of electrical synapses composed of Cx36 (gap junctions) on spinal cord anterior horn cells in mutant superoxide-dismutase 1 (SOD1) transgenic mice, which is a well-known ALS model. We also confirmed the finding in human autopsied spinal cord samples from ALS patients. Because electrical synapses composed of Cx36 are usually expressed by suppressive neurons, a reduction in neuronal Cx36 may lead to the hyperexcitability of anterior horn cells in motor neuron disease patients [80]. We also found that, in the progressive phase of SOD1 transgenic mice, astroglial Cx43 is increased while oligodendroglia Cx47 is decreased [81]. As astroglial Cx43 and oligodendroglial Cx47 form heterotypic gap junctions, an imbalance of Cxs in the pathological milieu indicates an increase in un-coupled astroglial Cx hemichannels. As astroglial Cx hemichannels release bioactive pro-inflammatory molecules, including potassium ions, ATP, and glutamate, and permit the entry of sodium and calcium ions, which could enhance astroglial activation, massive astrogliosis might play essential roles in the progressive phase of neurodegenerative diseases. The mechanism for hemichannel formation is currently under investigation. It is important to note that hemichannels may form due to the activation of astroglia, whether or not there is a decrease in oligodendroglial Cx47. Almad et al. also showed an increase in Cx43 in ALS patients and model mice. The study revealed a progressive increase in Cx43 expression in the SOD1_G93A_ mouse model of ALS and in ALS patients’ motor cortexes and spinal cords. This elevated Cx43 expression was also observed in astrocytes from SOD1_G93A_ mice and human-induced pluripotent stem cell-derived astrocytes. Increased Cx43 led to enhanced gap junction coupling, higher hemichannel-mediated activity, and elevated intracellular calcium levels in SOD1_G93A_ astrocytes compared to those in control astrocytes. Blocking Cx43, both pan Cx43 and Cx43 hemichannels, provided neuroprotection to motor neurons cultured with SOD1_G93A_ astrocytes, suggesting an unrecognized role of Cx43 in motor neuron loss associated with ALS [82,83]. Hashimoto et al. described that Cx30 was highly expressed in the pre-onset stage in mSOD1 mice and that Cx30 KO mice (Cx30KO-mSOD1 mice) showed delayed disease onset, while at the progressive and end stages of the disease, anterior horn cells were significantly preserved in Cx30KO-mSOD1 mice. Astrocyte activation was reduced in Cx30KO-mSOD1 mice compared with that in mSOD1 mice. Furthermore, the expression of Cx 43 in the pre-onset stage was downregulated in Cx30KO-mSOD1 mice, suggesting that the reduced expression of astroglial Cx30 in the early disease stage in ALS model mice protects neurons by attenuating astroglial inflammation [84].

### 4.5. Alzheimer’s Disease

Cx expression is increased in the brains of AD patients [85,86]. The amyloid precursor protein/presenilin-1 mouse model of AD also confirmed the increase in Cx. Neuronal Cx36 and Panx1 hemichannels are activated via the administration of amyloid Aβ_25–35_ peptide in vitro. The irreversible opening of these hemichannels is hypothesized to exert deleterious effects on neurons by inducing the nonspecific depolarization of neurons, possibly leading to cognitive decline [87]. Astroglial Cx43 and Cx30 are increased around amyloid plaques in AD model mice and human AD brains [85]. GFAP-positive astroglial Cx-ablated AD model mice show reduced neuronal damage, indicating pathogenic roles of astroglial Cxs [88]. A recent study has elucidated a feed-forward loop involving astroglial A2AR and Cx43 hemichannels, resulting in increased extracellular ATP-adenosine levels and the sustenance of enhanced astroglial A2AR activity in AD-like conditions [89].

### 4.6. Major Depressive Disorders and Autism Spectrum Disorders

Antidepressants exert different effects on Cx43 gap junctions and hemichannel activities. Some anti-depressants inhibit Cx43 hemichannels [90,91,92].

In postmortem brains of autism spectrum disorder (ASD) patients, Cx43 expression is increased in the superior frontal cortex. This increase in Cx43 also indicates a relationship between glial inflammation and ASD mechanisms [93]. Recently, interesting results have been published [94]. A study explored the role of astroglia in postnatal brain issues due to maternal inflammation, focusing on connexin hemichannels and pannexons. The opening of these channels can lead to [Ca^2+^] imbalance and astrocyte activation, which might harm astroglia survival and affect astrocyte-to-neuron support, making neurons more vulnerable to inflammation and subsequent immune challenges. [Ca^2+^] imbalances from these channels can harm astroglia survival and neuron support, making neurons more vulnerable to inflammation and subsequent immune challenges. Maternal inflammation can also trigger excitotoxicity through these channels, further harming neurons. Understanding hemichannels and pannexons in maternal inflammation-induced brain problems is vital for developing pharmacological therapies for affected offspring.

### 4.7. Brain Tumor

Brain metastasis from various cancers is a major cause of illness and death, often resistant to chemotherapy. Chen et al. explore the role of astroglia, the most abundant brain cells, in promoting brain metastasis. Breast and lung cancer cells express protocadherin 7 (PCDH7), facilitating the formation of gap junctions between cancer cells and astroglia using Cx43. This connection allows the transfer of cyclic guanosine monophosphate–adenosine monophosphate (cGAMP), a second messenger, to astroglia, triggering the STING pathway and the release of inflammatory cytokines. These cytokines, acting as paracrine signals, activate pathways in metastatic cells, promoting tumor growth and drug resistance. Meclofenamate and tonabersat, drugs modulating gap junctions, disrupt this signaling loop, offering a potential treatment for established brain metastasis [95]. Regarding primary CNS tumors, gap junctions interconnect glioblastoma cells, promoting long-distance communication and enabling the creation of pathways for brain microinvasion. The use of INI-0602, a gap junction blocker capable of crossing the blood–brain barrier, enhances the susceptibility of glioblastoma cells to the standard chemotherapy drug, temozolomide. This approach is linked to heightened JNK signaling pathway activity. The study by Potthoff et al. proposes that employing these inhibitors could present a promising therapeutic strategy in glioblastoma research [96]. Of note is that the bystander effect (BSE) also has a beneficial aspect of effectively delivering antitumor agents to tumor cells. Touraine et al. reported that functional GJ play a crucial role in the BSE and further support the notion that pharmacologically manipulating GJ might impact the outcomes of cancer therapy using hTK/ganciclovir [97].

## 5. Conclusions

In this review, we focus on the discovery, basic research, translational research, and clinical implications related to gap junctions and hemichannels, including Cxs and pannexins. Given the widespread presence of Cx/pannexin-expressing cells throughout the body, disturbances in homeostatic functions, whether congenital or acquired, can lead to various disorders. Open hemichannels release bioactive small molecules like ATP, adenosine, glutamate, and more, thereby activating neighboring cells and promoting inflammation. Consequently, hemichannel blockers are emerging as promising drug candidates.

Gap junctions within the CNS are crucial structures, enabling intercellular communication through Cx proteins. They go beyond mere structural components and play intricate roles in the pathogenesis of various neurological disorders. To understand their profound impact, we must delve into the complex web of cellular interactions within the CNS. Neurons and glial cells, including astroglia and oligodendrocytes, create an interconnected network through gap junctions, facilitating the rapid exchange of ions, metabolites, and signaling molecules. However, disruptions in this network can lead to severe CNS disorders, including epilepsy, multiple sclerosis, Alzheimer’s disease, autism spectrum disorders, ischemic brain lesions, demyelinating disorders, and more. Understanding gap junctions’ multifaceted role in these conditions is crucial for modern neuroscience. One promising avenue of research involves selectively modulating gap junction and hemichannel activity to develop novel drugs. Scientists aim to restore normal cell communication and inhibit excessive signaling, potentially revolutionizing the treatment of CNS disorders.

In conclusion, Cxs form complex communication channels in the CNS. Their study offers hope for innovative therapies and improved outcomes for those with CNS disorders.

## Figures and Tables

**Figure 1 ijms-24-16879-f001:**
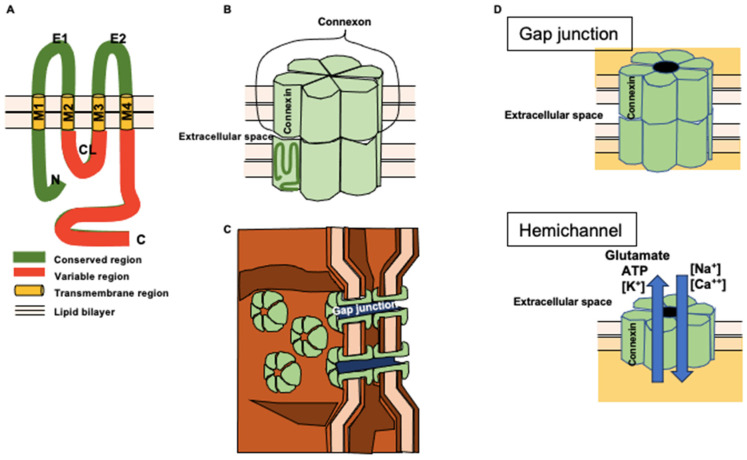
Schematic figures of Cx. (**A**) Ribbon molecular model of Cx. Cx comprising an N terminal region, four transmembrane regions (M1–4), two extracellular regions (E1; E2), a cytoplasmic loop (CL), and a C terminal region. Red ribbon indicates the variable regions, which characterize each Cx. (**B**) Gap junction assembled from Cxs. Green columns indicate each Cx. Six Cxs compose a connexon on the cell surface. Connexons form cell-to-cell gap junctions in a head-to-head configuration. (**C**) Cxs forming gap junctions by clustering on the cell surface. (**D**) A Cx-made gap junction comprising two connexons expressed on opposing cells. When connexons are expressed on the cell surface, they may work as hemichannels. Their opening leads to the uptake of Na^+^/Ca^2+^/Cl^−^, accompanied by the release of K^+^ and bioactive molecules such as ATP and glutamate. This may result in osmotic and ionic imbalances leading to cell death, and can provoke neuroinflammation.

**Figure 2 ijms-24-16879-f002:**
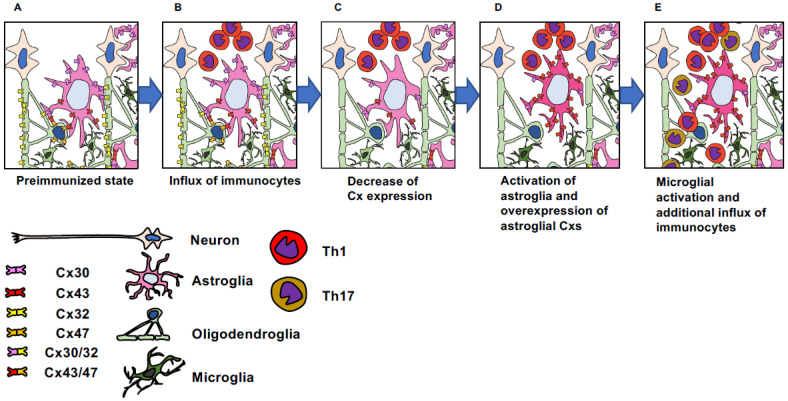
Cx imbalance induces CNS inflammation. (**A**) Each glial cell connected by Cxs except for microglia in the normal state. (**B**) Infiltration of the CNS parenchyma by autoreactive T cells when EAE is induced. (**C**) Decrease in Cxs on both astroglia and oligodendroglia in the acute stage of EAE. (**D**) Expression of an excess level of Cx43 by activated astroglia in the peak phase of EAE, while oligodendroglial Cx47 expression remains decreased. As a result, uncoupled Cx43 hemichannels are increased. Microglia are also activated. (**E**) Elevated levels of extracellular ATP, adenosine, and glutamate triggering the activation of other glial cells and endothelial cells, resulting in increased cytokines and chemokines. Consequently, this leads to a substantial influx of peripheral immune cells.

**Table 1 ijms-24-16879-t001:** Connexin expression in different cell types and related disorders.

Gene	Connexin (Human)	Expressed Cell Type	Relevance to Disease
*GJA1*	Cx43	Astroglia, Endothel, Heart, Fibroblast, T, B, Monocyte, Macrophage, Neutrophil, Dendritic Cell	ODDD (OMIM 164200)
*GJA3*	Cx46	Lung Epithelium, Eye Lens, Osteoblast	Cataract (OMIM 121015)
*GJA5*	Cx40	Endothel, Heart, T, B, Myoblast	Atrial Fibrillation (OMIM 121013)
*GJA8*	Cx50	Eye Lens	Cataracts (OMIM 600897)
*GJB1*	Cx32	Oligodendroglia, Schwann cells, Hepatocyte	CMTX1 (OMIM 302800)
*GJB2*	Cx26	Vestibular Epithelia, Hepatocyte, Kupffer Cell, Stomach, Keratinocyte, Lung Alveolar	Deafness (OMIM 601544)
*GJB3*	Cx31	Small Intestine, Colon, Skin	Deafness (OMIM 612644)
*GJB4*	Cx30.3	Keratinocyte	Skin disease (OMIM 605425)
*GJB5*	Cx31.1	Keratinocyte	Skin disease (OMIM 604493)
*GJB6*	Cx30	Astroglia, Keratinocyte	Deafness (OMIM 612643) Clouston syndrome (OMIM 129500)
*GJC2*	Cx47	Oligodendroglia	PMLD1 (OMIM 608804)

ODDD: Oculodentodigital dysplasia, CMTX1: Charcot–Marie–Tooth disease, X-linked 1, PMLD1: Pelizaeus–Merzbacher-like disease 1.

**Table 2 ijms-24-16879-t002:** Connexin expression in different cell types and related disorders.

Cells	Cxs	Manipulated Genotype	Effects of Each Connexin Deficiency on EAE		Possible Mechanisms	Refs.
			Acute EAE	Chronic EAE		
Astroglia	Cx30	KO	Unchanged	Attenuated	Induction of A2 astroglia Induction of M2-like microglia Downregulation of Cx43	[30]
	Cx43	icKO(GLAST-Cre)	Attenuated	Attenuated	Induction of A2 astroglia Suppression of immune cell infiltration Suppression of macrophage/microglia infiltration	[35]
	Cx30/Cx43	Cx30KO Cx43icKO(GFAP-Cre)	Unchanged	Unchanged	No change in BBB integrity, lesion load and pathology	[37]
Oligodendroglia	Cx47	icKO	Exacerbated	Exacerbated	Induction of A1 astroglia Promotion of microglia activation and macrophage infiltration Promotion of immune cell infiltration (Th17) Progressive demyelination	[38]
	Cx47	KO	Exacerbated	Exacerbated	Increase in the expression of Vcam-1, Ccl2, Gm-Cxsf BSCB disruption, enhanced gliosis, and increased T cell infiltration	[39]
	Cx32	KO	Exacerbated	Exacerbated	Disruption of gap junctions may induce the vulnerability of myelinated fibers	[29]
Others	Pannexin1	KO	Deleyed onset	Unchanged	Inhibition of ATP release and inflammasome activation	[41]

KO: knockout; icKO: inducible conditional knockout.

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
