# Peer review of "Connexins Control Glial Inflammation in Various Neurological Diseases"

_ijms, 2023, doi:10.3390/ijms242316879_

Round 1
Reviewer 1 Report
Comments and Suggestions for Authors
This manuscript attempts to review literature on the role of connexin-based channels in immune diseae, specifically on MS and the rodent EAE model. It incorrectly and neglectfully presents early studies, presents illogical hypotheses and even reports findings from studies in preparation.
Errors in fact include:
First cloning of connexin was by Kumar and Gilula and Paul in 1986, before Beyer published Cx43 and Nicholson Cx26.
Discovery of gap junction should not be attributed solely to JD Robertson, although his contribution was immense, see https://www.ncbi.nlm.nih.gov/pmc/articles/PMC2254793/
Figure 1 and statements throughout the text. Cytokines and chemokines are proteins with secondary and tertiary structure and molecular weights far in excess of the 1 kDa cutoff for permeability through connexin-based pores. It is difficult to understand how the concept of permeability to such large molecules has made it into the literature, but certainly should not be perpetuated.
There are numerous studies of gap junction/connexins in MS/EAE that are not cited in this overview and not included in Table 2; similarly, the pioneering studies by Celia Brosnan’s group on human and rodent MS/EAE are not mentioned.
Table 1 is not arranged alphabetically or by tissue or disease, and entries are questionable. For example, Cx32 expression in microglia is dubious, but abundance in liver is not cited; while Cx32 is present in oligodendrocytes, disease cited is most prominent in Schwann cells. Cx26 again is dubious in microglia; in which cells do mutations cause deafness?
L101. Cx29 has been speculated to work as a hemichannel, but “known to” vastly overstates it.
L114. While it is true that some neurons may express low levels of Cx45, this is much less prevalent than Cx36.
l.115. What does “Cx45 functions similarly to Cx36” mean? Channels are quite different in conductance and voltage dependence and most neurons have no Cx45.
L117. This interpretation of literature on microglial gap junctions is incorrect. The literature on gap junction expression in either resting or activated microglia is controversial, but the statement that certain connexins are expressed in one or another state is not established.
120. What does “together with pannexin” mean? Pannexin1 channels may well contribute much of the function attributed to connexin hemichannels, a viewpoint not considered in this review.
L120 and ff. Prominent missing citations are from groups of Faustmann and Saez and Brosnan.
L117. While it is correct that glia and neurons are not normally coupled, coupling does occur between developing neurons and glia and Cedric Raine reported it between axons and astrocytes in MS.
L124. How can https://doi.org/10.1002/glia.10141 and other papers not be cited here?
L137. The hypothesis that hemichannel abundance would increase if a neighboring cell did not produce a pairing connexon iso my knowledge without any supporting evidence whatsoever.
L140. What is the citation here?
L147. What does “therefore there is no phenotype” mean here?
Section 3.4 is not clear. Inflammation downregulates Cx43 in astrocytes, yet here it is stated that increased Cx43 hemichannels caused by imbalance of heterotypic gap junctions leads to increased cytokine release (how?). l204. What does this mean?
L217-257 describes data Nagata et al, Takase et al in preparation dealing with poorly characterized novel drugs (IGU and INI-0602). Non-peer-reviewed and unpublished results cannot be cited in a review article.
L273. No, GJB1 is not in astrocytes! And mutations span the entire coding region, not just Exon2!
L280. Malfunction is a vague term. Channel properties, trafficking, etc?dealing with gap juncitons in neurons and glia and seizure activity.
Section 4.2 is inadequate in describing the enormous literature
Section 4.3 should certainly include mention of bystander cell killing and gap junction involvement of spread from dying cells to neighbors.
L346. Unbalanced connexon expression is again hypothesized to lead to hemichannels without evidence, and that such hemichannels could release cytokines is fantasy.
Sections 4.5 and 4.6 are without rigorous citations, and the speculations that hemichannel opening leads to neuronal depolarization that causes cognitive decline and that Cx43 overexpression leads to ASD through increased inflammation are both preposterous as presented.
L370, What does “extend their significance far beyond surface observations” mean?
Comments on the Quality of English LanguageEnglish is satisfactory, although wordy and reducndant in places, such as Instroduction and conclusions
Author Response
Reviewer1
This manuscript attempts to review literature on the role of connexin-based channels in immune diseae, specifically on MS and the rodent EAE model. It incorrectly and neglectfully presents early studies, presents illogical hypotheses and even reports findings from studies in preparation.
Thank you very much for dedicating your valuable time to the thorough review. I truly appreciate your precise and invaluable feedback. It has come to my attention that my knowledge of connexins was outdated, and there were some misconceptions in my work. I have made numerous revisions and modifications based on your comments.
I kindly request that you review the revised manuscript once more and provide me with your feedback, whether it is favorable or unfavorable.
Errors in fact include:
First cloning of connexin was by Kumar and Gilula and Paul in 1986, before Beyer published Cx43 and Nicholson Cx26.
- I have modified the section (lines 27-30).
Discovery of gap junction should not be attributed solely to JD Robertson, although his contribution was immense, see https://www.ncbi.nlm.nih.gov/pmc/articles/PMC2254793/
- I have modified the section (lines 27-30).
Figure 1 and statements throughout the text. Cytokines and chemokines are proteins with secondary and tertiary structure and molecular weights far in excess of the 1 kDa cutoff for permeability through connexin-based pores. It is difficult to understand how the concept of permeability to such large molecules has made it into the literature, but certainly should not be perpetuated.
- Thank you for the indication. I have modified the concept that cytokines are sometimes released through the hemichannels(Figure 1, Line 56, 303-305).
There are numerous studies of gap junction/connexins in MS/EAE that are not cited in this overview and not included in Table 2; similarly, the pioneering studies by Celia Brosnan’s group on human and rodent MS/EAE are not mentioned.
- I have added other great studies of connexins on EAE (Lines 191-200, 210-237, Table 2).
[22] John, G. R.; Scemes, E.; Suadicani, S. O.; Liu, J. S.; Charles, P. C.; Lee, S. C.; Spray, D. C.; Brosnan, C. F., IL-1beta differentially regulates calcium wave propagation between primary human fetal astrocytes via pathways involving P2 receptors and gap junction channels. Proc Natl Acad Sci U S A 1999, 96, (20), 11613-8.
Table 1 is not arranged alphabetically or by tissue or disease, and entries are questionable. For example, Cx32 expression in microglia is dubious, but abundance in liver is not cited; while Cx32 is present in oligodendrocytes, disease cited is most prominent in Schwann cells. Cx26 again is dubious in microglia; in which cells do mutations cause deafness?
- I have made modifications to Table 1. In the Cx32 section, I have excluded microglia and included hepatocytes. I was unable to identify any Cx32-related human diseases apart from CMTX1. In the Cx26 section, I have omitted microglia. Cx26 is reported to be expressed on vestibular epithelial cells.
L101. Cx29 has been speculated to work as a hemichannel, but “known to” vastly overstates it.
-Thank you for the indication. I have amended the expression from "known to" to "reported" (Line 154).
L114. While it is true that some neurons may express low levels of Cx45, this is much less prevalent than Cx36.
- I have removed neuron from Cx45 section in Table 1.
l.115. What does “Cx45 functions similarly to Cx36” mean? Channels are quite different in conductance and voltage dependence and most neurons have no Cx45.
- I have deleted the part. Thank you for letting me know the fact.
L117. This interpretation of literature on microglial gap junctions is incorrect. The literature on gap junction expression in either resting or activated microglia is controversial, but the statement that certain connexins are expressed in one or another state is not established.
- I deleted the sentence: "Cx43 on microglia forms homotypic gap junctions."
- What does “together with pannexin” mean? Pannexin1 channels may well contribute much of the function attributed to connexin hemichannels, a viewpoint not considered in this review.
- Thank you for the suggestion. I have added a description of Pannexin1 in the following section. (Lines 217-237).
L120 and ff. Prominent missing citations are from groups of Faustmann and Saez and Brosnan.
- I'm sorry for the missing citation. I have added a description with indicated citation (Lines 124-131).
[22] John, G. R.; Scemes, E.; Suadicani, S. O.; Liu, J. S.; Charles, P. C.; Lee, S. C.; Spray, D. C.; Brosnan, C. F., IL-1beta differentially regulates calcium wave propagation between primary human fetal astrocytes via pathways involving P2 receptors and gap junction channels. Proc Natl Acad Sci U S A 1999, 96, (20), 11613-8.
[23] Faustmann, P. M.; Haase, C. G.; Romberg, S.; Hinkerohe, D.; Szlachta, D.; Smikalla, D.; Krause, D.; Dermietzel, R., Microglia activation influences dye coupling and Cx43 expression of the astrocytic network. Glia 2003, 42, (2), 101-8.
[24] Retamal, M. A.; Froger, N.; Palacios-Prado, N.; Ezan, P.; Saez, P. J.; Saez, J. C.; Giaume, C., Cx43 hemichannels and gap junction channels in astrocytes are regulated oppositely by proinflammatory cytokines released from activated microglia. J Neurosci 2007, 27, (50), 13781-92.
L117. While it is correct that glia and neurons are not normally coupled, coupling does occur between developing neurons and glia and Cedric Raine reported it between axons and astrocytes in MS.
- Thank you for the valuable information. I have added descriptions with citations (Lines 115-117).
[19] Soffer, D.; Raine, C. S., Morphologic analysis of axo-glial membrane specializations in the demyelinated central nervous system. Brain Res 1980, 186, (2), 301-13.
L124. How can https://doi.org/10.1002/glia.10141 and other papers not be cited here?
- I'm sorry for the under-citation in the previous manuscript. I have added a description with indicated citation (Lines 124-131).
[23] Faustmann, P. M.; Haase, C. G.; Romberg, S.; Hinkerohe, D.; Szlachta, D.; Smikalla, D.; Krause, D.; Dermietzel, R., Microglia activation influences dye coupling and Cx43 expression of the astrocytic network. Glia 2003, 42, (2), 101-8.
L137. The hypothesis that hemichannel abundance would increase if a neighboring cell did not produce a pairing connexon iso my knowledge without any supporting evidence whatsoever.
- Thank you for your feedback. The precise mechanism underlying the amelioration of EAE with Cx43 conditional ablation remains a subject of controversy. In acute MS lesions, both Cx43 and Cx47 levels decreased. However, in the chronic phase, Cx43 quickly recovered, accompanied by extensive astrogliosis, whereas Cx47 did not exhibit the same pattern. We are currently investigating the increase in hemichannel activity (Takase et al., submitted), although these findings have not been published yet.
We have added some speculation in section 4.4. (Lines 417-420)
L140. What is the citation here?
-I apologize for inadvertently including a different paper published by the same author. (Line 150)
[25] Markoullis, K.; Sargiannidou, I.; Gardner, C.; Hadjisavvas, A.; Reynolds, R.; Kleopa, K. A., Disruption of oligodendrocyte gap junctions in experimental autoimmune encephalomyelitis. Glia 2012, 60, (7), 1053-66.
L147. What does “therefore there is no phenotype” mean here?
-I'm sorry for the vague expression. I mean "there is no clinical phenotype"(Line 154).
Section 3.4 is not clear. Inflammation downregulates Cx43 in astrocytes, yet here it is stated that increased Cx43 hemichannels caused by imbalance of heterotypic gap junctions leads to increased cytokine release (how?).
- I apologize for any confusion caused by my previous statement. During the acute phase, there was a decrease in Cx43 expression, but in the chronic phase, as pronounced gliosis became more prominent in MS and EAE lesions, Cx43 quickly rebounded, and in some cases, even increased. This heightened expression of Cx43 hemichannels may manifest on the cell surface, leading to the release of ATP, which activates microglia. The activated microglia subsequently release proinflammatory cytokines such as IL-1b and TNF.
We are currently conducting in vivo and in vitro experiments to better understand the reasons behind the amelioration of glial inflammation due to Cx43 ablation by administrating INI-0602. While the ongoing results cannot be included here, I have introduced a speculative explanation in Section 4.4 (Lines 417-420).
L204. What does this mean?
- I meant that when one of the gap-junction-forming heterotypic connexons is removed, the other connexons may function as hemichannels, either opening or closing.
L217-257 describes data Nagata et al, Takase et al in preparation dealing with poorly characterized novel drugs (IGU and INI-0602). Non-peer-reviewed and unpublished results cannot be cited in a review article.
- I apologize for not being aware that we should not include unpublished data. I have removed the data description related to Dr. Takase's experiment. Dr. Nagata's paper has been published and cited (Line 275).
L273. No, GJB1 is not in astrocytes! And mutations span the entire coding region, not just Exon2!
-I have deleted the related part.
L280. Malfunction is a vague term. Channel properties, trafficking, etc?dealing with gap juncitons in neurons and glia and seizure activity.
- I have described ODDD in detail (Lines 319-326).
Section 4.2 is inadequate in describing the enormous literature
- I have added a description about Cxs and epilepsy (Lines 367-389).
Section 4.3 should certainly include mention of bystander cell killing and gap junction involvement of spread from dying cells to neighbors.
- I have added a description of the bystander effect and cited a related paper (Lines 399-401).
[56] Frantseva, M. V.; Kokarovtseva, L.; Perez Velazquez, J. L., Ischemia-induced brain damage depends on specific gap-junctional coupling. J Cereb Blood Flow Metab 2002, 22, (4), 453-62.
L346. Unbalanced connexon expression is again hypothesized to lead to hemichannels without evidence, and that such hemichannels could release cytokines is fantasy.
- Thank you for your valuable feedback. I wholeheartedly agree with your critique. I have removed the term "cytokine." The mechanism for hemichannel formation is currently under investigation. It's important to note that hemichannels may form due to the activation of astroglia, whether or not there is a decrease in oligodendroglial Cx47. I have included this speculation in the section (Lines 417-420).
Sections 4.5 and 4.6 are without rigorous citations, and the speculations that hemichannel opening leads to neuronal depolarization that causes cognitive decline and that Cx43 overexpression leads to ASD through increased inflammation are both preposterous as presented.
- I have added a description in detail (Lines 447-456).
L370, What does “extend their significance far beyond surface observations” mean?
- I have modified the conclusion section.
Dear reviewer,
Thank you again for your detailed feedback and valuable insights. Thanks to your input, our review has greatly improved. We are truly grateful. I have made revisions to the best of my ability, but please do not hesitate to point out any remaining deficiencies.
Sincerely,
Ryo Yamasaki
Reviewer 2 Report
Comments and Suggestions for Authors
Here is my review and critique of the manuscript:
The authors provide a comprehensive review of the roles of connexins and gap junctions in various neurological diseases. The introduction gives a good background on connexins, their structure and function in forming gap junctions for intercellular communication. The authors then nicely relate this to the importance of connexins and gap junctions between glial cells in the central nervous system.
The review covers relevant studies on connexins in both the peripheral and central nervous systems. For the central nervous system, the authors focus on astrocytes, oligodendrocytes and microglia, describing the connexins found in each cell type and their roles in forming homotypic and heterotypic gap junctions between the different glial cells. This provides important context for understanding how disrupting connexin expression or gap junction communication could contribute to pathology.
A major strength of the review is the section describing experimental autoimmune encephalomyelitis (EAE) studies in mice with conditional knockouts or deficiencies in connexins Cx30, Cx43 or Cx47. The authors clearly explain how loss of these connexins in astrocytes or oligodendrocytes leads to very different effects on EAE disease course and pathology. These studies provide compelling evidence for the significant yet distinct contributions of connexins in different glial cell types to demyelinating pathology.
The review of connexin-related heritable diseases, seizure disorders, ischemia, and neurodegenerative diseases is also quite informative, highlighting the clinical importance of gap junction communication. For each disease, the authors concisely summarize the relevant connexin affected and the resulting impact on gap junction function.
Overall, this is a well-written review that synthesizes findings from diverse studies to provide insight into the varied roles of glial connexins in neurological function and dysfunction. The authors have done a commendable job of explaining complex concepts clearly. The review is logically structured, flows well, and makes good use of summary tables and figures. I only have a few suggestions:
- The abstract could provide a bit more background in the introductory sentences to set the stage for readers unfamiliar with connexins.
- It would be helpful to define some abbreviations like CMTX1 and ODDD on first use.
- The conclusions could be expanded to highlight the most salient points and implications of the research covered.
But these are minor issues. Overall, I think this is a strong manuscript that makes a valuable contribution to the literature on glial connexins in neuropathology. The research presented significantly advances our understanding of how disrupting glial communication via connexins and gap junctions contributes to neurological disease.
Author Response
Reviewer2
Here is my review and critique of the manuscript:
The authors provide a comprehensive review of the roles of connexins and gap junctions in various neurological diseases. The introduction gives a good background on connexins, their structure and function in forming gap junctions for intercellular communication. The authors then nicely relate this to the importance of connexins and gap junctions between glial cells in the central nervous system.
The review covers relevant studies on connexins in both the peripheral and central nervous systems. For the central nervous system, the authors focus on astrocytes, oligodendrocytes and microglia, describing the connexins found in each cell type and their roles in forming homotypic and heterotypic gap junctions between the different glial cells. This provides important context for understanding how disrupting connexin expression or gap junction communication could contribute to pathology.
A major strength of the review is the section describing experimental autoimmune encephalomyelitis (EAE) studies in mice with conditional knockouts or deficiencies in connexins Cx30, Cx43 or Cx47. The authors clearly explain how loss of these connexins in astrocytes or oligodendrocytes leads to very different effects on EAE disease course and pathology. These studies provide compelling evidence for the significant yet distinct contributions of connexins in different glial cell types to demyelinating pathology.
The review of connexin-related heritable diseases, seizure disorders, ischemia, and neurodegenerative diseases is also quite informative, highlighting the clinical importance of gap junction communication. For each disease, the authors concisely summarize the relevant connexin affected and the resulting impact on gap junction function.
Overall, this is a well-written review that synthesizes findings from diverse studies to provide insight into the varied roles of glial connexins in neurological function and dysfunction. The authors have done a commendable job of explaining complex concepts clearly. The review is logically structured, flows well, and makes good use of summary tables and figures. I only have a few suggestions:
- The abstract could provide a bit more background in the introductory sentences to set the stage for readers unfamiliar with connexins.
- I have revised the introduction section to encompass a broader range of connexin functions (Lines 8-22).
- It would be helpful to define some abbreviations like CMTX1 and ODDD on first use.
- I have spelled out the abbreviations. Thank you.
- The conclusions could be expanded to highlight the most salient points and implications of the research covered.
- I have re-wrote the conclusion section (Lines 476-500).
But these are minor issues. Overall, I think this is a strong manuscript that makes a valuable contribution to the literature on glial connexins in neuropathology. The research presented significantly advances our understanding of how disrupting glial communication via connexins and gap junctions contributes to neurological disease.
-
Dear reviewer,
Thank you for your detailed review of our manuscript. We appreciate your positive feedback and insights.
Your comments on the clarity of the introduction and the coverage of connexins in glial cells are encouraging. The distinction between astrocytic and oligodendrocytic connexins in EAE is a key focus, and we're glad it came across effectively.
Your acknowledgment of the clinical relevance of connexin-related diseases and your positive evaluation of the review's structure are much appreciated. We aimed to provide a clear and well-organized overview.
We value your input, which has been instrumental in enhancing our work. If you have further comments or questions, please feel free to contact us.
Sincerely,
Ryo Yamasaki.
Reviewer 3 Report
Comments and Suggestions for Authors
In this review, Dr. Yamasaki has done an extensive and well described work on the role of connexins in physiological function and disease in the brain.
The review is well written and the large amount of aspects covered by the manuscript makes it of interest for a large audience.
I recommend the author to include a small chapter (4.7 for example) on brain cancer. I can propose to the author the following fundamental papers on the topic that mention the importance of gap junction and connexins in the progression of brain tumor: Chen et al 2016 (Carcinoma-astrocyte gap junctions promote brain metastasis by cGAMP transfer); Ratto et al. 2020 (Squaring the Circle: A New Study of Inward and Outward-Rectifying Potassium Currents in U251 GBM Cells); Patthoff et al. 2019 (Inhibition of Gap Junctions Sensitizes Primary Glioblastoma Cells for Temozolomide).
Best Regards,
Author Response
Reviewer3
In this review, Dr. Yamasaki has done an extensive and well described work on the role of connexins in physiological function and disease in the brain.
The review is well written and the large amount of aspects covered by the manuscript makes it of interest for a large audience.
I recommend the author to include a small chapter (4.7 for example) on brain cancer. I can propose to the author the following fundamental papers on the topic that mention the importance of gap junction and connexins in the progression of brain tumor: Chen et al 2016 (Carcinoma-astrocyte gap junctions promote brain metastasis by cGAMP transfer); Ratto et al. 2020 (Squaring the Circle: A New Study of Inward and Outward-Rectifying Potassium Currents in U251 GBM Cells); Patthoff et al. 2019 (Inhibition of Gap Junctions Sensitizes Primary Glioblastoma Cells for Temozolomide).
- Thank you for the suggestion. I have added a new section as your recommendation (Lines 458-474).
Dear reviewer,
Thank you for your kind words regarding our review on connexins in brain function and disease. We genuinely appreciate your feedback and your insightful suggestion to include a chapter on brain cancer.
If you have any further recommendations or ideas, please feel free to share them. Your expertise is greatly valued.
Sincerely,
Ryo Yamasaki
Reviewer 4 Report
Comments and Suggestions for Authors
The manuscript is of limited novelty and should be rejected. Abstract, Table 1 and Table 2 are (almost) identical to the published paper by the same author in Clin Exp Neuroimmunol. 2020;11 (Suppl. 1):34–40 DOI: 10.1111/cen3.12568.
Several key references are missing, e.g. regarding Cx43 hemichannels in ALS the papers by Almad et al. 2016, 2022 (PMID: 27083773, PMID: 35312356).
Comments on the Quality of English Languageok
Author Response
Reviewer4
The manuscript is of limited novelty and should be rejected. Abstract, Table 1 and Table 2 are (almost) identical to the published paper by the same author in Clin Exp Neuroimmunol. 2020;11 (Suppl. 1):34–40 DOI: 10.1111/cen3.12568.
-Thank you for the suggestion. I have modified the Abstract, and tables.
Several key references are missing, e.g. regarding Cx43 hemichannels in ALS the papers by Almad et al. 2016, 2022 (PMID: 27083773, PMID: 35312356).
-Thank you for the valuable comments. I have added citations and descriptions (Lines 417-429).
[60] Almad, A. A.; Doreswamy, A.; Gross, S. K.; Richard, J. P.; Huo, Y.; Haughey, N.; Maragakis, N. J., Connexin 43 in astrocytes contributes to motor neuron toxicity in amyotrophic lateral sclerosis. Glia 2016, 64, (7), 1154-69.
[61] Almad, A. A.; Taga, A.; Joseph, J.; Gross, S. K.; Welsh, C.; Patankar, A.; Richard, J. P.; Rust, K.; Pokharel, A.; Plott, C.; Lillo, M.; Dastgheyb, R.; Eggan, K.; Haughey, N.; Contreras, J. E.; Maragakis, N. J., Cx43 hemichannels contribute to astrocyte-mediated toxicity in sporadic and familial ALS. Proc Natl Acad Sci U S A 2022, 119, (13), e2107391119.
Dear Reviewer,
Thank you for your valuable feedback. I have made the necessary revisions and incorporated recent findings regarding connexins and associated disorders. I kindly request you to review our updated manuscript, and I welcome any further input or suggestions you may have.
Sincerely,
Ryo Yamasaki
Round 2
Reviewer 1 Report
Comments and Suggestions for Authors
This revision of a submission with numerous concerns has been improved in some respects but serious issues remain.
Title: What does “Gap junction rules glial inflammation” mean? By rule, is control or regulate or dominate or dictate or determine meant? By gap junction, are both GJ and HCs meant? If so, perhaps connexins should be used. [eg connexins control/determine glial inflammation…”]
8 Connexins do not facilitate GJ formation, they underlie it.
L12 Triggered HC opening remains an hypothesis
L14 What evidence is there for the statement that channel size depends on cellular context?
L15, Wht does typically mean? Is there any sold evidence that larger molecules permeate?
L19-20 HC involvement is hypothesized, not proven
L28. NO, the discovery of connexins should be credited to David Paul (reference provided in my previous review) or by Paul and Kumar/Gilula reference you added.
L55. MAY work as HCs
l68,69 and ff. Ionic charges should be superscripted, Ca is divalent
Taqble 1. Last column heading should read relevance to disease not of disease. OMIMs without disease associations (eg, GJA9,10, GJB7, GJC1) provide no disease relevance and should be deleted
L106. NO Cx32 is not in peripheral myelin. Rather, it connects cytoplasmic compartments that have been squeezed off by the myelin
L103. NO Cx29 has been proposed to form HCs; there is no strong evidence that it does so
L118 Cx36
L120 Cx43 is proposed to form HCs
L124 Reference for reduction of Cx43 in coculture?
L129 missing reference
L134. Meaning? That there is no clinical phenotype associated with loss or mutation of Cx30?
L152-172. This is unclear and rambling and should be focused and clarified.
L192 and many other occurences. Delete previously, as every study cited here was published previously
L210 and highlighted text needs to be edited
L220 reference?
L226 reference for statement beginning “..when we used mefloquine..”?
L244 What is the evidence that such imbalance leads to excessive Cx43 HCs or that they are functional?
L249 What is the evidence for this statement: “When one gap junction-forming connexin is ablated, other connexins become closed hemichannels”?
L287, 295 what previous reports?
L310-2 combine sentences
L319-326. This summary of how GJ mutations might affect function belongs earlier, either as overview on this topic or after considering CMTX, where there are hundreds of mutatikons, so that such a conclusion can actually be reached.
L388 Since Cx32 is expressed primarily in oligodendrocytes, this is an opportunity to comment on relevance of myelinating cells to neuronal excitability
L463 what is cGAMP? cAMP? cGMP?
L470 Note that on l284 this compound was touted as being HC specific, but now it is proposed as a GJ blocker?
L480 What does the statement that HCs result from “inadequate opening of cell surface connexons” mean?
Comments on the Quality of English Language
All revised sections should be checked carefully for English usage.
Author Response
Reviewer1
This revision of a submission with numerous concerns has been improved in some respects but serious issues remain.
Title: What does “Gap junction rules glial inflammation” mean? By rule, is control or regulate or dominate or dictate or determine meant? By gap junction, are both GJ and HCs meant? If so, perhaps connexins should be used. [eg connexins control/determine glial inflammation…”]
- I have modified the title. [Line 2]
L8 Connexins do not facilitate GJ formation, they underlie it.
- I have rephrased "facilitate" into "forms". [Line 8]
L12 Triggered HC opening remains an hypothesis
- I have rephrased "can" into "assumed to". [Line 12]
L14 What evidence is there for the statement that channel size depends on cellular context?
- Reviewer 2 also raised the concern about the text. I have rephrased as suggested by reviewer 2: " The size of the channel pore is depending on the connexin isoform and cellular context-specific effects such as posttranslational modifications ". [Lines 13-15]
L15, Wht does typically mean? Is there any sold evidence that larger molecules permeate?
- I have deleted the word "typically ". [Line 15]
L19-20 HC involvement is hypothesized, not proven
- I have rephrased the sentence as "Hemichannels are hypothesized to contribute to proinflammatory effects by releasing ATP, adenosine, glutamate, and other bioactive molecules, leading to neuroglial inflammation." [Lines 20-22]
L28. NO, the discovery of connexins should be credited to David Paul (reference provided in my previous review) or by Paul and Kumar/Gilula reference you added.
- I have added the sentence: " The exploration of gap junctions began with their discovery by Revel and Karnovsky {Revel, 1967}, and cloned by Kumar {Kumar, 1986} and Paul {Paul, 1986},".[Lines 29-30]
L55. MAY work as HCs
- I have amended it as so. [Line 57]
l68,69 and ff. Ionic charges should be superscripted, Ca is divalent
- I have amended it as so. [Lines 71, 72, etc.]
Taqble 1. Last column heading should read relevance to disease not of disease. OMIMs without disease associations (eg, GJA9,10, GJB7, GJC1) provide no disease relevance and should be deleted
- Thank you for the suggestions. I have amended the points. [Table 1]
L106. NO Cx32 is not in peripheral myelin. Rather, it connects cytoplasmic compartments that have been squeezed off by the myelin
- I have amended it as " Cx32 is expressed by Schwann cells and is thought to be important for the proliferation of Schwann cells, re-myelination, and nerve regeneration".[Lines 93-94]
L103. NO Cx29 has been proposed to form HCs; there is no strong evidence that it does so
- I have deleted Cx29 from the text.
L118 Cx36
- Fixed. Thank you.[Line 119]
L120 Cx43 is proposed to form HCs
- Fixed. Thank you.[Line 120]
L124 Reference for reduction of Cx43 in coculture?
- Ref. 27(Watanabe, M.; Masaki, K.; Yamasaki, R.; Kawanokuchi, J.; Takeuchi, H.; Matsushita, T.; Suzumura, A.; Kira, J. I., Th1 cells downregulate connexin 43 gap junctions in astrocytes via microglial activation. Sci Rep 2016, 6, 38387)
In the referenced paper, we confirmed the reduction of Cx43 on astroglia in mixed glial cell culture, which included astroglia and microglia, upon the addition of Th1 (IFN-gamma) conditioned media.
L129 missing reference
- I have added Refs 27 and 28. Thank you. [Line 129]
Ref. 27 Watanabe, M.; Masaki, K.; Yamasaki, R.; Kawanokuchi, J.; Takeuchi, H.; Matsushita, T.; Suzumura, A.; Kira, J. I., Th1 cells downregulate connexin 43 gap junctions in astrocytes via microglial activation. Sci Rep 2016, 6, 38387.
Ref. 28 John, G. R.; Scemes, E.; Suadicani, S. O.; Liu, J. S.; Charles, P. C.; Lee, S. C.; Spray, D. C.; Brosnan, C. F., IL-1beta differentially regulates calcium wave propagation between primary human fetal astrocytes via pathways involving P2 receptors and gap junction channels. Proc Natl Acad Sci U S A 1999, 96, (20), 11613-8.
L134. Meaning? That there is no clinical phenotype associated with loss or mutation of Cx30?
- I have removed the sentence" In the pre-immunized state of EAE, Cx30 is barely expressed by glial cells; therefore, there is no clinical phenotype." as it might confuse readers. Instead, I have rewritten the paragraph. [Lines 158-174]
L152-172. This is unclear and rambling and should be focused and clarified.
- Sorry for my poor English. I have rewritten the part. [Lines 158-174]
L192 and many other occurences. Delete previously, as every study cited here was published previously
- I have deleted "previous" or "previously" as much as possible. Thank you.
L210 and highlighted text needs to be edited
-Fixed. Thank you. [Lines 212-218]
L220 reference?
- Inserted. Thank you. [Line 225]
Ref. 44 Lutz, S. E.; Gonzalez-Fernandez, E.; Ventura, J. C.; Perez-Samartin, A.; Tarassishin, L.; Negoro, H.; Patel, N. K.; Suadicani, S. O.; Lee, S. C.; Matute, C.; Scemes, E., Contribution of pannexin1 to experimental autoimmune encephalomyelitis. PLoS One 2013, 8, (6), e66657.
L226 reference for statement beginning “..when we used mefloquine..”?
- Fixed. Thank you. [Line 229]
L244 What is the evidence that such imbalance leads to excessive Cx43 HCs or that they are functional?
- Because we can't label CxHCs, we could not confirm the increase of Cx43HCs in Cx47 deficient EAE. We are now trying to check the functional activation of primary-cultured astroglial HCs after the "A1" activation cytokine cocktail as mentioned in the previous edition(Ezgi, in preparation). There is no clear evidence, and this is just speculation, which is discussed in the referred paper. (Ref. 41 Zhao, Y.; Yamasaki, R.; Yamaguchi, H.; Nagata, S.; Une, H.; Cui, Y.; Masaki, K.; Nakamuta, Y.; Iinuma, K.; Watanabe, M.; Matsushita, T.; Isobe, N.; Kira, J. I., Oligodendroglial connexin 47 regulates neuroinflammation upon autoimmune demyelination in a novel mouse model of multiple sclerosis. Proc Natl Acad Sci U S A 2020, 117, (4), 2160-2169.) I agree with your comments and amended the expression of the part by inserting the phrase "It is hypothesized". [Line 247]
L249 What is the evidence for this statement: “When one gap junction-forming connexin is ablated, other connexins become closed hemichannels”?
- There is no clear evidence, and this is just speculation, which is discussed in the referred paper. (Zhao, Y.; Yamasaki, R.; Yamaguchi, H.; Nagata, S.; Une, H.; Cui, Y.; Masaki, K.; Nakamuta, Y.; Iinuma, K.; Watanabe, M.; Matsushita, T.; Isobe, N.; Kira, J. I., Oligodendroglial connexin 47 regulates neuroinflammation upon autoimmune demyelination in a novel mouse model of multiple sclerosis. Proc Natl Acad Sci U S A 2020, 117, (4), 2160-2169.)
I agree with your comments and amended the expression of the part by inserting the phrase "We hypothesized that ~ might". [Lines 252-254]
L287, 295 what previous reports?
- I have inserted a reference. [Lines 300, 306]
Ref. 48 Takeuchi, H.; Mizoguchi, H.; Doi, Y.; Jin, S.; Noda, M.; Liang, J.; Li, H.; Zhou, Y.; Mori, R.; Yasuoka, S.; Li, E.; Parajuli, B.; Kawanokuchi, J.; Sonobe, Y.; Sato, J.; Yamanaka, K.; Sobue, G.; Mizuno, T.; Suzumura, A., Blockade of gap junction hemichannel suppresses disease progression in mouse models of amyotrophic lateral sclerosis and Alzheimer's disease. PLoS One 2011, 6, (6), e21108.
L310-2 combine sentences
- Combined into one sentence. Thank you. [Lines 329-332]
L319-326. This summary of how GJ mutations might affect function belongs earlier, either as overview on this topic or after considering CMTX, where there are hundreds of mutatikons, so that such a conclusion can actually be reached.
- I agree. I have moved the section to the top of the paragraph. Thank you. [Lines 318-326]
L388 Since Cx32 is expressed primarily in oligodendrocytes, this is an opportunity to comment on relevance of myelinating cells to neuronal excitability
- Thank you. I have added a more detailed summary of the reference paper. [Lines 405-415]
L463 what is cGAMP? cAMP? cGMP?
- cyclic guanosine monophosphate–adenosine monophosphate (cGAMP). [Lines 506-507]
L470 Note that on l284 this compound was touted as being HC specific, but now it is proposed as a GJ blocker?
- Fixed as hemichannel blocker. Thank you. [Line 514]
L480 What does the statement that HCs result from “inadequate opening of cell surface connexons” mean?
- I removed "inadequate". Thank you. [Line 524]
Reviewer 3 Report
Comments and Suggestions for Authors
The paper has been improved and it is suitable for publication.
best regards
Author Response
Reviewer3
The paper has been improved and it is suitable for publication.
- Thank you for taking the time to review the revision.
Reviewer 4 Report
Comments and Suggestions for Authors
Please find my comments attached.

Comments on the Quality of English LanguageThe quality of English is sufficient.
Author Response
Reviewer4
The manuscript has been improved, however, there are still many issues that need to be addressed. Please carefully revise the manuscript according to the below comments.
l.13 remove „in one cell“ since hemichannels rarely open in one cell only
- Fixed. Thank you. [Line 12]
l.14 “The size of the channel depends on the specific connexin structure and the cellular context.”
This sentence is very vague, I would suggest to reword as follows: The size of the channel pore is depending on the connexin isoform and cellular context-specific effects such as posttranslational modifications.
- Thank you for the suggestion. I have replaced the sentence as suggested. [Lines 13-15]
l.14-16 Hemichannels allow various bioactive molecules, typically under 1000 kDa, to move in and out of the host cell in the direction of the electrochemical gradient.
Please add the underlined text. This information is vital to understand the direction of flow via an open hemichannel (also see below comments).
- I have added the part. Thank you. [Line 16]
l.36 “connexins have a relatively short half-life of 1 to 2.5 hours”
Since ref 6 (PMID: 7287816) states five-hour half-life of mouse liver gap-junction protein, I would suggest to reword the sentence as follows: connexins have a relatively short half-life of 1 to 5 hours.
- Fixed. Thank you. [Line 38]
Fig. 1D and legend l.55 “When connexons are expressed on the cell surface, they work as hemichannels, which release bioactive molecules such as ATP, glutamate, and ions.”
Opening of hemichannels allows the flow of ions or metabolites in the direction of their electrochemical gradient, meaning that Na+ and Ca++ can enter the cell while K+ can be released. Thus the arrow in Fig. 1D indicating ion movements should point in both directions.
- Fixed. Thank you for your valuable comment. [Figure 1D]
l.69 correct typo H+/Ca+, Calcium is a divalent ion (Ca++ or Ca2+)
- Fixed. Thank you. [Line 72 etc.]
l.120 “Cx43 also forms hemichannels, together with pannexin, to release ATP, Ca2+, and glutamate [1].”
The statement that Ca2+ is released via open Cx43 hemichannels is not supported by reference [1] and is clearly wrong under physiological conditions where there is a high extracellular concentration of Ca2+ (1-2 mM range) and very low intracellular Ca2+ concentration (~100 nM in the resting state and ~ 1-10 uM in excited cells), which means that the direction of the electrochemical gradient for Ca2+ is inwards. Schalper et al. have demonstrated that Cx43 hemichannels facilitate Ca2+ entry (PMID: 20881238).
- Fixed and added the reference. Thank you.[Lines 120-121]
Ref. 26 Schalper, K. A.; Sanchez, H. A.; Lee, S. C.; Altenberg, G. A.; Nathanson, M. H.; Saez, J. C., Connexin 43 hemichannels mediate the Ca2+ influx induced by extracellular alkalinization. Am J Physiol Cell Physiol 2010, 299, (6), C1504-15.
l.124 “Interferon gamma that released from Th1 cells were one of the cause of microglial activation [21].” Suggested rewording: Interferon gamma released from Th1 cells was suggested to induce microglial activation [21].
- Fixed as suggested. Thank you. [Lines 125-126]
l.129 “Microglia were reported to Cx43 gap junction mediated dye coupling [23].” This sentence is incomplete, suggested rewording: Microglia were reported to modulate Cx43 gap junction-mediated dye coupling [23].”
- Fixed as suggested. Thank you. [Lines 129-130]
l.130/131 “The decrease of gap junction-forming Cx43 and increase of Cx43 hemichannels on astroglia by inflammatory cytokines also may potentiate the chronic brain inflammation [24].”
Indeed, De Bock et al. have shown that Cx43 hemichannel opening in astrocytes was induced by IL-6 in a calcium- dependent manner and that LPS-triggered astrogliosis could be prevented by Tat-Gap19, a selective Cx43- hemichannel inhibitor, or BAPTA-AM which suppressed Cx43 hemichannel opening (PMID: 35881483).
- Thank you for letting me know about the newest paper about the Cx43 HC opening in astroglia. I have added it as suggested. [Lines 132-137]
l.133-136 Figure 2 should be discussed in more detail in the main text. The link between MCT1 and Glucose uptake on the one hand, and AQP4 and K+ shuttling on the other hand is not described in the text, these transport processes should be introduced together with the metabolic function of connexins and the relevant references should be included. The monocarboxylate transporter 1 (MCT1) is known to be involved in lactate transport, while members of the glucose transporter family GLUT facilitate glucose uptake (PMID: 29490274, PMID: 32789766). The mechanistic link between K+ uptake and AQP4 water permeability has been subject of discussion (PMID: 23277478).
- Thank you for your detailed advice. I have added as recommended. [Lines 110-113]
l.196 “We also could not detect any significant difference in EAE clinical severity in Cx30-deficient mice in the acute phase but found the mitigation of clinical, pathological, and genetic phenotypes in the chronic phase of EAE.” Please include your reference at the end of this sentence.
- Thank you. I have revised the paragraph according to the query from Reviewer 1. I have inserted our paper about Cx30KO EAE. [Lines 158-174]
l.210 Stavropoulos et al. also published an excellent paper
- Fixed. Thank you. [Lines 212]
l.226 “Furthermore, when we used mefloquine (MFQ), a pharmacological inhibitor of Panx1 channels” Does the “we” refer to the author’s research or to the findings by Lutz et al.?
- Fixed. Thank you. [Line 229]
l.249 “When one gap junction-forming connexin is ablated, other connexins become closed hemichannels” Please include references here to support this statement or put forward as a hypothesis.
- I have amended it as " We hypothesized that when one gap junction-forming connexin is ablated, other connexins might become closed hemichannels." [Lines 252-254]
l.259 “As mentioned in the excellent studies from other groups” Please include references here and refer to Table 2 if necessary.
- I have added refs and added mention to table 2. Thank you. [Table 2]
l.297 Figure 3 should be introduced in the main text of the manuscript.
Regarding connexin imbalance and CNS inflammation, De Bock et al. (PMID: 35881483) recently demonstrated that LPS and proinflammatory cytokines trigger endothelial hemichannel opening while IL-6 activates astrocytic hemichannels, thus targeting connexin hemichannels is a promising therapeutic approach to suppress barrier leakage as well as astrogliosis. Delvaeye et al. (PMID: 31719598) have previously shown that blocking Cx43 hemichannels with TAT-Gap19 protected mice against TNF-induced mortality, hypothermia and vascular leakage in the course of an acute cytokine storm.
- Figure 3 is mentioned in Lines 263-269.
I have added suggested text. Thank you.
l.318 ODDD mutations affect Cx43 trafficking as well as channel functions, for review see section VII in PMID: 28931622. The ODDD Cx43 mutants G138R and G60S have been characterized to display increased hemichannel function combined with decreased gap junction function. The I130T mutant has been linked to ventricular tachyarrhythmia in a mouse model (PMID: 18077386).
- Thank you very much for the useful information. [Lines 344-347]
l.369 Administration of the selective Cx43-hemichannel inhibitor Tat-Gap19 has been shown to decrease seizure activity in animal epilepsy models, indicating involvement of Cx43 hemichannels in epilepsy (PMID: 29683209).
- Thank you for the useful comment. I have added the suggested information. [Lines 388-390]
l.390 Regarding role of Cx43 hemichannels in brain ischemia, Freitas-Andrade et al. (PMID: 30872361) have demonstrated that pharmacological blockade of Cx43 hemichannels with TAT-Gap19 significantly decreased infarct volume in an animal stroke model, the detrimental hemichannel activity following ischemic stroke was linked to MAPK phosphorylation of sites at the Cx43 C-terminus. Chen et al. (PMID: 30386214) found that Gap19 exhibited neuroprotective effects on cerebral ischemia/reperfusion and that Gap19 reduced the inflammatory response via inhibition of the TLR4 signaling pathway following in vivo middle cerebral artery occlusion and in vitro oxygen glucose deprivation.
- Thank you for the information. I have added it as suggested. [Lines 428-435]
l.415 “Because astroglial Cx hemichannels release bioactive pro-inflammatory molecules, including ions, adenosine triphosphates (ATPs), and glutamates, massive astrogliosis might play essential roles in the progressive phase of neurodegenerative diseases.”
As explained above, hemichannels facilitate the entry of Na+ and Ca2+ into the cell and the escape of K+, ATP, and other molecules in the direction of their electrochemical gradient. Thus, this sentence should be reworded accordingly.
-Thank you. I have fixed as follows: "As astroglial connexin (Cx) hemichannels release bioactive pro-inflammatory molecules, including potassium ions, ATP, and glutamate, and permit the entry of sodium and calcium ions, which could enhance astroglial activation, massive astrogliosis might play essential roles in the progressive phase of neurodegenerative diseases." [Lines 449-452]
l.429 Hashimoto et al. described that Cx30 was highly expressed in the pre-onset stage in mSOD1 mice and that Cx30 KO mice (Cx30KO-mSOD1 mice) showed delayed disease onset, while at the progressive and end stages of the disease anterior horn cells were significantly preserved in Cx30KO-mSOD1 mice. Astrocyte activation was reduced in Cx30KO-mSOD1 mice compared with mSOD1 mice. Furthermore, expression of connexin 43 at the pre-onset stage was downregulated in Cx30KO-mSOD1 mice, suggesting that reduced expression of astroglial Cx30 at the early disease stage in ALS model mice protects neurons by attenuating astroglial inflammation.
- Thank you for the kind consideration. I added our results in the section. [Lines 464-471]
l.434 “The irreversible opening of these hemichannels exerts deleterious effects on neurons by inducing nonspecific depolarization of neurons, leading to cognitive decline.”
Please include references here to support this statement or put forward as a hypothesis. A direct link between hemichannel opening, neuronal depolarization and cognitive decline in patients has not been established so far.
- Thank you for the question. Referenced review by Koulakoff (PMID 22008509) introduced Orellana's work (PMID 21451035). After their in vivo and in vitro work, they concluded as " Thus, Abeta leads to a cascade of hemichannel activation in which microglia promote the release of glutamate and ATP through glial (microglia and astrocytes) hemichannels that induces neuronal death by triggering hemichannels in neurons." As you mentioned, the effects of HC opening on patients' cognitive decline are not confirmed, yet. I have added Orellana's paper as a reference and added a statement as hypothetic. [Lines 476-477]
l.447 Recently, interesting results has have been published [70].
- Fixed. Thank you. [Line 491]
l.451 [Ca2+]i imbalances linked to the opening of these channels
- Fixed. Thank you. [Line 494]
Round 3
Reviewer 1 Report
Comments and Suggestions for Authors
This manuscript is further improved, but issues remain, as detailed below.
As a general comment, I personally find the evidence for pathological HC involvement in most cases rather shaky, and the speculated "imbalance" between synthesis of connexins destined to be cojoined leading to overabundance of open HCs seems pretty far-fetched. But with adequate qualifiers such as "hypothesized" and "speculated", it is OK to present the ideas for scientific debate. The authors (and readers) should be wary of such labels as "selective hemichannel inhibitor" attached to Gap19 or INI-0602, where efficacy of blockade is in many cases modest and not totally selective.
Specific comments:
l4. form
l12. change "assumed" to "are believed"
l13. delete membrane
l14, change is depending to depends
l30 Kumar and Gilula
l31 Beyer
table 1: I previously commented that OMIM references for connexins without known disease does not make sense for column headed role in disease. This has not been changed
l110-113. Delete as not relevant
l121: connexins do not form HC together with pannexin; rephrase.
l329: this sentence has become garbled when combined; the simple point is that CMTX is caused by Cx32 mutations
l518. Bystander cell killing through gap junction mediated exchange of gancyclovir metabolites is the opposite of what is proposed here and is worth a comment
Comments on the Quality of English Language
Flowery phrases (eg,line 83) should be toned down throughout; describing a previous study as excellent is not necessarily accurate or appropriate and such adjectives should be removed.
Minor tense and syntax errors should be corrected.
Author Response
This manuscript is further improved, but issues remain, as detailed below.
As a general comment, I personally find the evidence for pathological HC involvement in most cases rather shaky, and the speculated "imbalance" between synthesis of connexins destined to be cojoined leading to overabundance of open HCs seems pretty far-fetched. But with adequate qualifiers such as "hypothesized" and "speculated", it is OK to present the ideas for scientific debate. The authors (and readers) should be wary of such labels as "selective hemichannel inhibitor" attached to Gap19 or INI-0602, where efficacy of blockade is in many cases modest and not totally selective.
-Your detailed review is greatly appreciated. It raises important points about uncertain evidence linking pathological hemichannel involvement and potential imbalances in connexin synthesis. Your emphasis on using qualifiers like "hypothesized" and "speculated" in scientific discussion is invaluable for presenting ideas rigorously.
Additionally, your careful advice about labeling compounds such as Gap19 or INI-0602 as "selective hemichannel inhibitors," considering their limited effectiveness and lack of absolute selectivity, is noteworthy in scientific discussions. Notably, INI-0602 binds to the outer loop of Cxs, inhibiting GJ/HC opening and causing irreversible internalization, effectively blocking GJ/HC.
Your insights greatly contribute to a more nuanced and critical understanding of these concepts. Thank you for your thorough and thought-provoking evaluation.
Specific comments:
l4. form
- Fixed. (Line 8)
l12. change "assumed" to "are believed"
- Fixed. (Line 12)
l13. delete membrane
- Fixed.
l14, change is depending to depends
- Fixed. (Line 14)
l30 Kumar and Gilula
- Fixed. (Line 30)
l31 Beyer
- Fixed. (Line 31)
table 1: I previously commented that OMIM references for connexins without known disease does not make sense for column headed role in disease. This has not been changed
-Sorry for my misunderstanding. I have deleted OMIM121012, 611925, 607058, 607425, 611922, 611925 from the table 1.
l110-113. Delete as not relevant
-Thank you. Deleted.
l121: connexins do not form HC together with pannexin; rephrase.
- I have deleted "pannexin". (Line 117)
l329: this sentence has become garbled when combined; the simple point is that CMTX is caused by Cx32 mutations
- I have deleted "garbled" part from the sentence. (Lines 323-324)
l518. Bystander cell killing through gap junction mediated exchange of gancyclovir metabolites is the opposite of what is proposed here and is worth a comment
- I added the following text: "Meanwhile, the bystander effect (BSE) also has a beneficial aspect of effectively delivering antitumor agents to tumor cells. They reported that functional GJ plays a crucial role in the BSE and further supports the notion that pharmacologically manipulating GJ might impact the outcomes of cancer therapy using hTK/ganciclovir {Touraine, 1998}." (Lines 523-526)
Reviewer 4 Report
Comments and Suggestions for Authors
Please find my comments attached.

Comments on the Quality of English LanguageModerate editing of English is required as indicated in the attached document.
Author Response
Reviewer 4
Unfortunately, the manuscript still contains many mistakes and imperfections, moreover, several key references are absent and should be added in the context of this review. For all these reasons, major revision is again required before the paper can be considered for publication. Please find a detailed list of comments below.
Major comments:
L.57/58 When connexons are expressed on the cell surface, they may work as hemichannels. Their opening leads to the uptake of Na+/Ca2+/Cl-, accompanied by the release of K+ and bioactive molecules such as ATP and glutamate. This may result in osmotic and ionic imbalances leading to cell death, and can provoke neuroinflammation.
- Thank you for the amendments. (Line 55-59)
Figure 2 is misleading and should be redone. It is wrongly stated that glucose is taken up from the blood into astroglia via MCT1 and that K+ permeates through AQP4. As mentioned in the previous review round, MCT1 is a transporter not for glucose but for monocarboxylates which include lactate, pyruvate and the ketone bodies B-hydroxybutyrate and acetoacetate (see PMID: 8124722, MID: 32144120). Glucose is transported into cells via several glucose transporters from the GLUT family (MID: 8240230) or via Na+-dependent transporters. The predominant GLUT transporter proteins involved in cerebral glucose utilization are GLUT1 and GLUT3, with GLUT1 being present in brain astrocytes and oligodendrocytes while neurons use GLUT3 (see e.g. MID: 26528968 for review). Regarding K+ uptake from neurons into oligodendroglia, it needs to be considered that K+ is excluded from the water-selective AQP4 pore and that AQP4 could affect K+ conductance indirectly via modulation of the inward-rectifying potassium channel Kir4.1 (MID: 24137016).
Oligodendrocytes express Kir4.1, a channel well suited to participate in extracellular K+ clearance (MID: 29596047, MID: 33763430). At the other end, axonally-derived K+ and associated osmotic water are expelled by Kir4.1 and AQP4 in astrocyte endfeet that surround capillaries (PMID: 19850107).
- Thank you for your advice. I agree with your suggestion. Consequently, I have removed Figure 2 and descriptions related to glucose or K+ influx because the homeostatic function of connexins is beyond the scope of this review.
Consistent referencing is required in Chapter 4.
Chapter 4.1 Heritable diseases caused by connexin gene mutations: Here it should be added that connexin mutations can cause a gain-of-function effect by increasing the permeability of hemichannels, leading to "leaky" hemichannels and cell death (PMID: 25398718).
- I have added the following sentence: "Connexin mutations can cause a gain-of function effect by increasing the permeability of hemichannels, leading to "leaky" hemichannels and cell death{Kelly, 2015 #136}." (Lines 319-320)
In 4.1.1 please include the study by Liang et al. reporting about a case of severe neuropathy with leaky Cx32 hemichannels
(PMID: 15852376). While localization and trafficking of the mutant protein in cell culture was normal, electrophysiological studies revealed that the involved Cx32 missense mutation F235C caused aberrant hemichannel opening with excessive membrane permeability and adverse effects on Schwann cell viability.
- I have added as recommended. (Lines 327-330)
Chapter 4.1.4: Hearing loss in syndromic deafness likely differs from the nonsyndromic form with mutant hemichannels that are overactive and/or show altered Ca2+ permeability leading to cell dysfunction and death (PMID: 28778872).
- I have added as recommended. (Lines 362-364)
Chapter 4.1.5: Levit et al. reviewed the role of hemichannels in the pathogenesis of Keratitis-Ichthyosis-Deafness (KID) syndrome associated with C×26 mutation (PMID: 21933663). Overactive, "leaky" hemichannels may deplete the cytoplasm of essential metabolites, depolarize the plasma membrane, induce apoptosis via Ca2+ overload, or cause lysis via osmotic imbalance.
- I have added as recommended. (Lines 376-380)
L.399 Panx1 is a protein that forms channels similar to connexin-based gap junctions: This statement is incorrect, since pannexins bear significant sequence homology with the invertebrate gap junction proteins, innexins and more distant similarities in their membrane topologies and pharmacological sensitivities with the gap junction proteins, connexins, however, unlike the connexin gap junction channels, pannexins have in the majority of studies been shown to be functional in single membranes, without forming intercellular channels (PMID: 21532340).
- Thank you for the advice. I have removed the wrong description. (Line 401)
L.426-8 This suggests that gap-junctional communication contributes to ischemic injury propagation (so called "bystander effects", and targeting specific gap junctions may reduce brain damage [66]. However, it has been reported that gap junctional communication may be neuroprotective in ischemia and that C×43 heterozygous null mice exhibited a significantly larger infarct volume compared to wild type (MID: 12064612)
- Thank you for the advice. I have added following description: "On the contrary, Naus et al. reported that blocking gap junctions during a glutamate insult to co-cultures of astroglia and neurons results in increased neuronal injury. They hypothesized that gap junctions play a neuroprotective role against glutamate toxicity{Naus, 2001 #141}. The effects of blocking gap junctions in the context of an ischemic insult may vary depending on the pathological circumstances." (Lines 436-440)
Minor comments:
Moderate editing of English is required as indicated below:
L.8 Use the plural form: Connexins (Cs) form gap junctions
- I have replaced all "connexins" into "Cs".
L.12 Changes in the intracellular or extracellular environment are assumed to trigger
- I have replaced the word "assume" into "are believed" according to the advice from reviewer 1. (Line 12)
l.28 Gap junctions, assembled from connexins
- Fixed. (Line 28)
L.30 cloning by Kumar [3] and Paul [4]
- Fixed included Reviewer1's comment. (Line 30)
L.214 The gene names should be written in italics.
- Fixed. (Line 206)
L.250 such as ATPs or glutamate
- Fixed. (Line 243)
L.263 Figure 3
- Fixed. (Line 256)
L.272 Table 2
- Fixed. (Line 265)
L.316 Connexins involved in other neuronal diseases
- Fixed.
L.352 PMLD1 cases
- Fixed.
L.405 A study examined
- Fixed.
L.447 Cx47
- Fixed.
L.503 This study: which study is meant here?
- Fixed.
L.542 In conclusion, connexins form complex communication channels in the CNS
- Fixed.
Round 4
Reviewer 4 Report
Comments and Suggestions for Authors
Please find my comments attached.

Comments on the Quality of English LanguageMinor editing of English language is required as indicated in the attached document.
Author Response
Revision Round 4
Reviewer 4
The manuscript has been significantly improved, however, a few issues still need to be addressed. I suggest that the manuscript should be accepted after minor revision. Please amend the manuscript according to the comments below.
For connexin please use the abbreviation Cx throughout the manuscript text, as listed in the Abbreviations.
- Thank you. I used Cx for connexin, and Cxs for connexins.
l.5/6 In the affiliation, remove the “and” before Japan, a comma should be placed before the country name: Kyushu University, Japan
- Thank you. Fixed. (Line 5)
l.15 Insert the correct unit in the sentence: Hemichannels allow various bioactive molecules, under ~1 kDa
- Thank you. Fixed. (Line 15)
l.29-32 Citations are mixed up, the sentence should read: The exploration of gap junctions began with their discovery by Revel and Karnovsky [2], and cloning by Kumar and Gilula [3], and Paul [4], followed by the cloning and sequencing of connexin 43 (Cx43), a 43-kilodalton (kDa) protein, by Beyer [5] and Cx26 by Zhang and Nicholson [6].
- Thank you. Fixed. (Lines 30)
l.32/33 Use the scientific term “connexin isoforms” instead of “connexin family members”: Humans have a family of connexin genes, and most cell types express multiple connexin isoforms.
- Thank you. Fixed. (Line 33)
l.51/52 Use the plural form in the sentence: Red-ribbon indicates the variable regions, which characterize each connexin.
- Thank you. Fixed. (Line 51)
l.199 in contrast to the phenotype
- Thank you. Fixed. (Line 198)
l.234 Since the following paragraph is concerned with gap junctions and hemichannels, I would advise to change the title to “Tuning of the brain inflammatory milieu by alteration of connexin channel functions”
- Thank you. Fixed. (Line 233)
l.243 ATP or glutamate
- Thank you. Fixed. (Line 241)
l.319 Moreover, connexin mutations can cause a gain-of function effect by increasing the permeability of hemichannels, leading to "leaky" hemichannels and cell death [46].
- Thank you. Fixed. (Line 318)
l.323 Charcot–Marie–Tooth disease, X-linked 1 (CMTX1), is caused by a mutation
- Thank you. Fixed. (Line 322)
l.326 At the end of this sentence, please add the missing reference by Jeng LJ, Balice-Gordon RJ, Messing A, Fischbeck KH, Scherer SS. The effects of a dominant connexin32 mutant in myelinating Schwann cells. Mol Cell Neurosci.2006; 32: 283–98
- Thank you. I have inserted the ref as indicated. (Line 329)
l.347 Insert blank: PMLD1 cases
- Thank you. Fixed. (Line 342)
l.353-361 and l.366-376 Please provide more references for the interested reader!
- Thank you for the advice. I have inserted the sentence: "It is also well-known that a single mutation, 35delG, is responsible for the majority of cases of autosomal recessive hearing loss known as DFNB1. This specific gene mutation stands as the most prevalent cause of hearing loss within American and European populations, boasting a carrier rate of approximately 3%. Additionally, the mutation 167delG, another deletion variant causing DFNB1, holds a carrier rate of around 4% within the Ashkenazi-Jewish population {Cohn, 1999 #143}." (Lines 363-368)
- I have also added references about KID syndrome (Skinner 1981, Shanker 2012, Montgomery 2004, Caceres-Rios 1996, van Steensel 2002). (Lines 371-381)
l.401 Panx1: Panx1 is a protein that forms pannexon, also known as pannexin 1 channel [60].
- Thank you. Fixed. (Line 405)
l.406 Remove blank in Cx32: Cx32 deficiency leads to neuronal hyperexcitability
- Thank you. Fixed. (Line 411)
l.421 “abnormal milieu” sounds very vague and is not mentioned in the quoted reference [66]. I would suggest to reword the sentence as follows: Usually, connexin hemichannels are closed, but they can open under pathological conditions [66].
- Thank you. Fixed. (Lines 426-427)
l.443 The sentence should probably read: Neuronal synapses comprise not only of the metabolic synapses but also of electrical synapses [72].
- Thank you. Fixed. (Line 448)
l.483 possibly leading to cognitive decline [80].
- Thank you. Fixed. (Line 487)
l.487 ATP-adenosine levels
- Thank you. Fixed. (Line 492)
l.497 Suggested rewording: The study explored the role of astrocytes in postnatal brain issues due to maternal inflammation, focusing on connexin hemichannels and pannexons. Opening of these channels can lead to [Ca2+]i imbalance and astrocyte activation which might harm astroglia survival and affect astrocyte-to-neuron support, making neurons more vulnerable to inflammation and subsequent immune challenges.
- Thank you. Fixed. (Lines 500-504)
l.508 Chen et al. explore the role of astroglia
- Thank you. Fixed. (Line 513)
l.521 The study by Potthoff et al. proposes that employing these inhibitors could present a promising therapeutic strategy in glioblastoma research [89].
- Thank you. Fixed. (Line 527)
l.523 Of note, the bystander effect (BSE) also has a
- Thank you. Fixed. (Line 528)
l.524 Touraine et al. reported that that functional GJ play a crucial role in the BSE
- Thank you. Fixed. (Line 529)
l.533 Open hemichannels release bioactive small molecules like ATP, adenosine, glutamate, and more,
- Thank you. Fixed. (Line 538)